# MathReal: We Keep It Real! A Real Scene Benchmark for Evaluating Math Reasoning in Multimodal Large Language Models

## Abstract

Multimodal Large Language Models (MLLMs) have demonstrated remarkable capabilities in visual mathematical reasoning across various existing benchmarks. However, these benchmarks are predominantly based on clean or processed multimodal inputs, without incorporating the images provided by real-world Kindergarten through 12th grade (K–12) educational users. To address this gap, we introduce MathReal, a meticulously curated dataset comprising 2,000 mathematical questions with images captured by handheld mobile devices in authentic scenarios. Each question is an image, containing the question text and visual element. We systematically classify the real images into three primary categories: image quality degradation, perspective variation, and irrelevant content interference, which are further delineated into 14 subcategories. Additionally, MathReal spans five core knowledge and ability categories, which encompass three question types and are divided into three difficulty levels. To comprehensively evaluate the multimodal mathematical reasoning abilities of state-of-the-art MLLMs in real-world scenarios, we design six experimental settings that enable a systematic analysis of their performance. Through extensive experimentation, we find that the problem-solving abilities of existing MLLMs are significantly challenged in realistic educational contexts. Based on this, we conduct a thorough analysis of their performance and error patterns, providing insights into their recognition, comprehension, and reasoning capabilities, and outlining directions for future improvements.

## 1 Introduction

Recent advances in Large Language Models (LLMs) have catalyzed the development of MLLMs, which are capable of jointly interpreting visual and textual information. This evolution has substantially enhanced model performance across a broad range of multimodal understanding tasks, including visual question answering, diagram interpretation, document analysis, and mathematical reasoning. As MLLMs become increasingly adept at bridging text and vision, their reasoning capabilities, particularly in domains requiring precise symbol processing and structured logic, have drawn significant attention from the research community.

With the rapid development of reasoning models, an increasing number of mathematical reasoning benchmarks have been proposed, including both pure-text benchmarks and multimodal benchmarks. Pure-text mathematical reasoning benchmarks, such as AIME24 Ankner et al. (2024), AIME25 Jaech et al. (2024), OlympiadBench He et al. (2024), and Polymath Wang et al. (2025e), primarily focus on evaluating reasoning ability from textual question statements. More recently, multimodal benchmarks have been introduced to incorporate visual contexts, such as MathVista Lu et al. (2023), MathVerse Zhang et al. (2024b), TrustGeoGen Fu et al. (2025), MM-MATH Sun et al. (2024), MathVision Awais et al. (2024), LogicVista Xiao et al. (2024), DynaMath Zou et al. (2024), VisOnlyQA Kamoi et al. (2024), MathGlance Sun et al. (2025), VisioMath Li et al. (2025), MV-MATH Wang et al. (2025b), GeoEval Zhang et al. (2024a), and We-Math Qiao et al. (2024). These benchmarks provide diverse evaluation settings that test not only pure symbolic reasoning but also multimodal perception and reasoning, thereby driving progress in the development of more general and robust MLLMs.

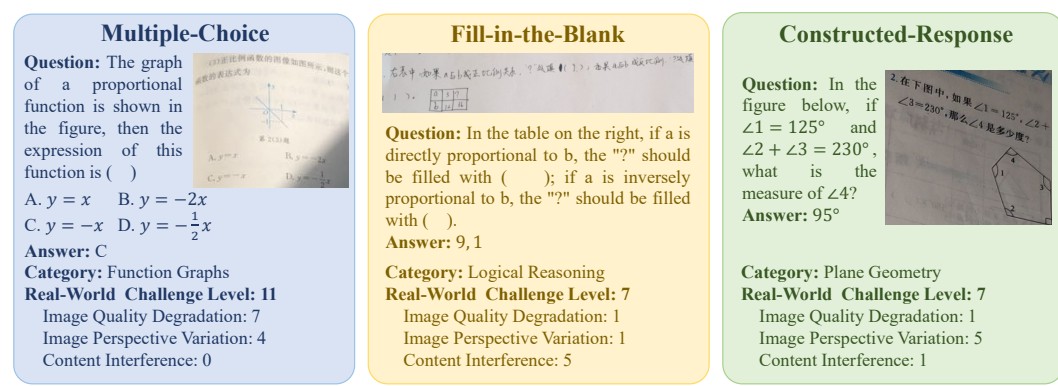

Figure 1: Sampled MATHREAL examples from each question type. Each question contains a real image and annotated information.

Despite these advancements, the majority of existing multimodal math benchmarks consist of clean or post-processed images, which rarely account for cases encountered by real-world users, making it difficult to assess how multimodal models perform in real environments. For instance, K–12 users often capture textbook pages or homework questions using handheld mobile devices to ask models for help. Real-world scenarios are often more challenging than traditional clean image inputs and the entire question text is embedded within the image, unlike conventional benchmarks that frequently rely on textual inputs. Additionally, mathematical question images captured by real-world users often reflect a distribution that differs substantially from both prior multimodal math benchmarks and the training data of existing models, as they are embedded in authentic educational contexts and aligned with real user needs, thereby posing joint challenges for both perception and reasoning.

To bridge this gap, we introduce MATHREAL, a novel benchmark designed to assess the performance of MLLMs on real-world, visually grounded K–12 mathematical questions. To support this, we develop a comprehensive data construction pipeline tailored to real-world multimodal math questions, addressing the challenges of collection, annotation, and validation under realistic conditions. MATHREAL comprises 2,000 high-quality questions sourced from authentic educational contexts, each captured via mobile photography as an image containing a figure, requiring models to first perceive visual content before performing reasoning. We define three primary challenges commonly encountered in real-world K–12 educational scenarios: *image quality degradation*, *perspective variation*, and *irrelevant content interference*, which are further divided into 14 fine-grained subcategories, such as *blur*, *rotation*, *handwritten answers*, etc.

To evaluate the multimodal mathematical reasoning abilities of MLLMs under real-world conditions, we construct MATHREAL with carefully designed annotations. Every question image spans five core knowledge and ability categories, three question types, and three difficulty levels. The dataset includes three question types and is systematically categorized across three difficulty levels and five knowledge domains, such as geometry, algebra, statistics, logical reasoning, and function graphs. To ensure high-quality and consistent annotations, each question is independently verified by at least two expert annotators, and is enriched with precise ground-truth metadata, including the ground-truth question text, detailed descriptions of visual elements, and correct answers.

We conduct extensive evaluations on MATHREAL across 4 LLMs and 40 multimodal models. Even in relatively simple K–12 scenarios, the best-performing model Doubao-1.5-thinking-vision-pro attains only 53.9% accuracy, in sharp contrast to the near-human or competition-level performance often reported on established mathematical benchmarks, underscoring the substantial gap to real-world applicability and the necessity of MATHREAL grounded in authentic educational scenarios. In conclusion, the contributions of this paper are summarized as follows:

- We propose MATHREAL, the first real-world benchmark of 2,000 K–12 multimodal math questions photographed in natural settings, covering 3 systematic characterizations of real-world scenarios, 5 knowledge and ability categories, 3 question types, and 3 difficulty levels.

Table 1: Key Statistics of MATHREAL. The unit of question length is words.

| Statistic | Number |
|---|---|
| Total questions | 2000 |
| - Multiple-Choice Questions | 104 |
| - Fill-in-the-Blank Questions | 475 |
| - Constructed-Response Questions | 1421 |
| Questions in the testmini set | 480 |
| Elementary-level Questions | 779 |
| Middle School-level Questions | 883 |
| High School-level Questions | 338 |
| Questions with only real images | 745 |
| Questions with real images and clean images | 1255 |
| Questions with a single figure | 1296 |
| Questions with multiple figures | 704 |
| Questions with a single sub-question | 829 |
| Questions with multiple sub-questions | 1171 |
| Minimum question length | 7 |
| Maximum question length | 451 |
| Average question length | 122.03 |
| Average answer length | 27.25 |

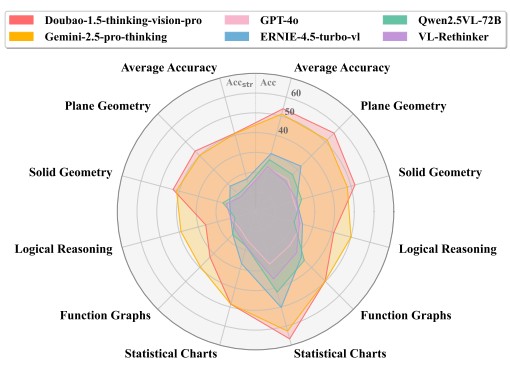

Figure 2: Performance comparison of six MLLMs on five categories and overall average accuracy. The radar chart shows results under two evaluation standards: strict accuracy ($Acc_{str}$) and loose accuracy (Acc), symmetrically arranged across 12 axes.

- We evaluate 40 MLLMs under 6 experimental settings to assess their reasoning abilities under real-world conditions. Our results demonstrate a notable performance gap between real and clean images, indicating that existing MLLMs remain far from reliable when applied in real-world educational scenarios.

- Through controlled experiments, we demonstrate that visual conditions commonly encountered in real-world scenarios, such as blur, rotation, and handwritten answers, significantly impair the reasoning performance of current MLLMs. In contrast, these models achieve notably higher accuracy when provided with clean textual or visual inputs, indicating that their visual perception components remain fragile when exposed to realistic distortions.

## 2 MATHREAL

While MLLMs have shown strong performance on existing visual math benchmarks, these benchmarks predominantly feature clean inputs and rarely reflect usage in real-world educational scenarios. This is particularly relevant because MLLMs have the potential to explain solutions and evaluate answer correctness in real educational settings. To bridge this gap, we present MATHREAL, a benchmark grounded in naturally captured images and designed to evaluate MLLMs under realistic visual conditions.

### 2.1 REAL VISUAL MATH DATASET

**Dataset Overview.** MATHREAL comprises 2,000 math question instances, each represented as a noisy image captured via handheld mobile devices under real conditions. All images are sourced from authentic K–12 educational materials, including textbooks, exam papers, and printed exercises. The photographs reflect a wide range of real-world acquisition scenarios, encompassing three major categories of noise: image quality degradation, image perspective variation, and handwriting interference. These three categories are further divided into a total of 14 fine-grained subtypes, providing a rich taxonomy of real-world imperfections. This collection process intentionally preserves the complexity and imperfection inherent to mobile-based image capture in practical settings.

Each sample in MATHREAL is an image that contains a complete math question, with both the question text and the associated figures embedded within the image rather than provided as separate clean inputs. The dataset includes 1,296 questions with a single figure and 704 questions with multiple figures. It also includes 829 questions with a single sub-question and 1,171 with multiple sub-questions, providing diverse reasoning structures. All questions are manually annotated with three supplementary elements: the ground-truth question text (QG), an exact visual description of the

figure present in the image (DG), and the correct reference answer. The purpose of these annotations is to enable a systematic analysis of models' multimodal perception and reasoning abilities in real-world scenarios.

The dataset includes three types of questions: multiple-choice, fill-in-the-blank, and constructed-response. In terms of academic stage, questions are distributed across three educational stages: primary school, middle school, and high school, ensuring coverage of content across the K–12 spectrum. Additionally, 745 questions are accompanied only by real images, while 1,255 are paired with both real images and clean images, which exclude real-world artifacts. The dataset also includes a testmini subset of 480 questions. Detailed statistics on question types and visual content categories are summarized in Table 1.

**Data Collection Process.** We construct the dataset by sampling 1.5 million photographed math questions from a large-scale user-uploaded repository. A two-stage filtering process is applied to ensure quality and relevance. First, a domain-specific classifier selects math-related samples containing figures. Then, GPT-4o, Doubao-1.5-vision-pro-32k, and Qwen2.5-VL-Instruct-72B independently evaluate each image to determine whether it contains a single, complete question and whether the figure is essential. Samples with irrelevant visuals or dialogue-style formats are excluded. Only those approved by all three models are retained, resulting in a high-quality dataset for evaluating the visual reasoning capabilities of MLLMs.

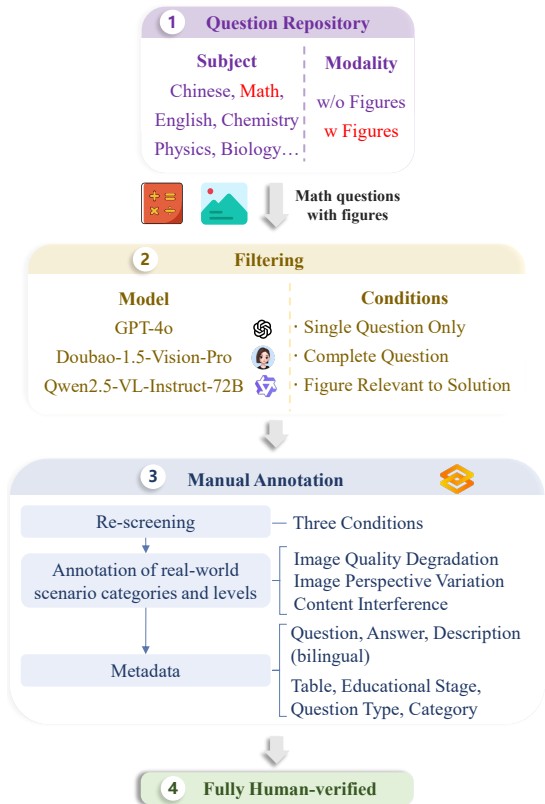

Figure 3: The flowchart of data construction, including data filtering and manual annotation.

**Data Annotation Process.** We build a Gradio-based platform and organize the annotation into three fully manual stages. In Stage One, we filter out samples that do not meet benchmark criteria, such as incomplete questions, multiple-question images, or irrelevant figures. In Stage Two, we annotate image conditions according to a predefined taxonomy covering three major real-world scenario types. In Stage Three, we annotate question-level metadata, including question content, type, educational stage, knowledge category, figure descriptions, and ground truth answers. All question-level metadata annotations (including real-world challenge level) are conducted independently by two different professional annotators. In cases where the two annotators disagree, a third professional annotator will re-annotate the sample until consensus is reached. Detailed annotation rules for the real-world challenge level are provided in the Appendix. In the end, we conduct a fully human-verified process to ensure that the final dataset reflects diverse real-world conditions while maintaining high semantic and structural quality for evaluating multimodal models.

## 2.2 DATA CHARACTERISTICS

In contrast to other MLLMs math reasoning datasets, the unique characteristics of MATHREAL are summarized as "vision-only input" and "in-the-wild challenges". These two features better align with the data distribution in real educational scenarios and pose distinct challenges to the perception and reasoning capabilities of MLLMs.

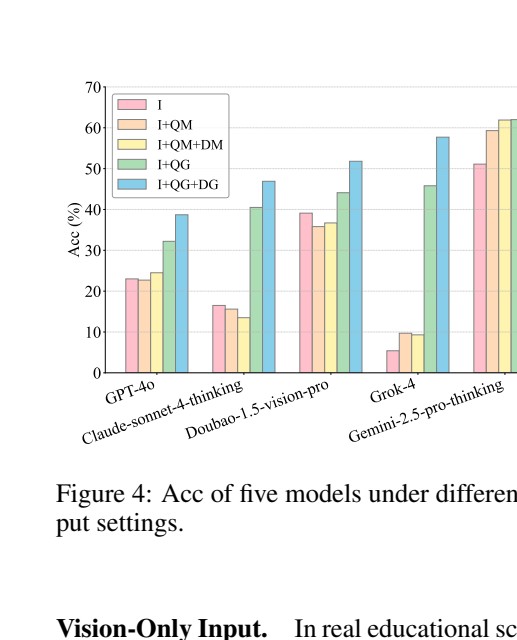

Figure 4: Acc of five models under different input settings.

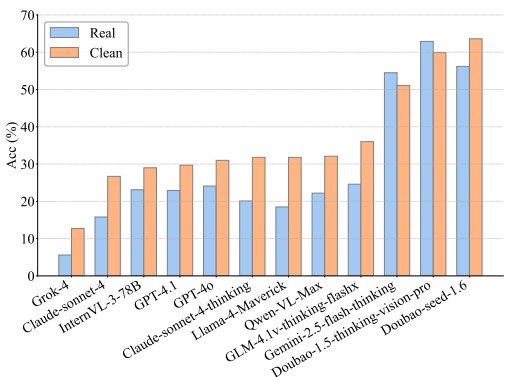

Figure 5: Acc comparison of models on real vs. clean images across selected 175 samples in MATHREAL *testmini*.

**Vision-Only Input.** In real educational scenarios, all information necessary for solving mathematical questions, including the question statement, figures, or diagrams, is typically contained within a single image. This requires models to first perceive and extract key information from the image before proceeding to reason and solve the question. Correspondingly, MATHREAL uses a single raw image as the sole input. However, to decouple perception and reasoning, the dataset provides QG and DG as supplementary annotations , facilitating fine-grained evaluation of MLLMs' capabilities.

**In-the-Wild Challenges.** In real educational scenarios, raw images often contain substantial noise due to unconstrained capture conditions. This challenges models to robustly perceive critical content while ignoring non-essential artifacts. To reflect this realism, MATHREAL categorizes noise into three major categories, encompassing 14 fine-grained subtypes. Specifically, image quality degradation includes blur, underexposure/overexposure, shadow coverage, and glare; image perspective variation includes rotation, in-plane tilt, non-planar capture, and background distortion; irrelevant content interference includes handwritten questions, reverse side content, question marking, figure marking, handwritten answer for multiple-choice questions, and handwritten process for constructed-response questions. Detailed annotations are provided for each subtype.

## 3 EXPERIMENT

### 3.1 EXPERIMENTAL SETUP

**Data Preparation and Subset Division.** The MATHREAL dataset contains 2,000 questions. To enable faster evaluation and model development validation, we divide the dataset into two subsets: *testmini* and *test*. The *testmini* subset includes 480 questions and serves as a validation set for model development or for users with limited computational resources. The *test* subset consists of the remaining 1,520 questions and functions as the standard evaluation set. We use a stratified random sampling strategy across different categories, ensuring that the sample sizes within each stratum are proportional to those in the full dataset, thus maintaining statistical representativeness. In the experiments that follow, all quantitative results are reported using the *testmini* subset of MATHREAL.

**Experimental Settings.** To evaluate the reasoning capability of MLLMs in real-world, image-based mathematical questions, we design six experimental settings that progressively disentangle visual perception and reasoning. Each question is an image containing both textual content (the *question*) and visual elements (the *figure*, which can be represented by a textual *description*). Based on this, three primary input modalities are defined: image only (I), which serves as the primary evaluation; image with human-annotated question text (I+QG); and image with human-annotated question text and figure description (I+QG+DG). Two reasoning paradigms are considered: a *one-stage* approach, where the model performs question recognition and reasoning jointly from the raw image ($I_{UER}$), and a *two-stage* approach, where the model first generates intermediate

Table 2: Comparison of model performances across five categories. PG: Plane Geometry, SG: Solid Geometry, LR: Logical Reasoning, FG: Function Graphs, SC: Statistical Charts. $Acc_{str}$ is strict accuracy, Acc is loose accuracy. The **first** and second highest accuracy of LLMs are bolded and underlined, respectively.

| Model | $Acc_{str}$ | | | | | | Acc | | | | | |
|---|---|---|---|---|---|---|---|---|---|---|---|---|
| | PG | SG | LR | FG | SC | Avg | PG | SG | LR | FG | SC | Avg |
| *LLMs* (Question Text + Figure Description, CoT with 0-shot) | | | | | | | | | | | | |
| Qwen3-235B-A22B-thinking | 29.1 | 30.6 | 41.3 | 20.9 | 48.5 | 31.2 | 35.2 | 36.4 | 48.8 | 27.5 | 61.4 | 37.9 |
| DeepSeek-V3 | 27.5 | 31.5 | 34.8 | 27.9 | 57.6 | 31.2 | 42.4 | 36.5 | 46.9 | 41.3 | 69.9 | 43.3 |
| Qwen3-235B-A22B-instruct | 34.0 | 33.3 | 37.0 | 39.5 | 45.5 | 35.4 | 46.0 | 40.9 | 50.8 | 52.3 | 60.5 | 46.8 |
| DeepSeek-R1 | 42.9 | 36.9 | 41.3 | 30.2 | 57.6 | 41.2 | 56.3 | 44.5 | 51.7 | 47.7 | 77.0 | 53.8 |
| *Closed Models* (Image-only, CoT with 0-shot) | | | | | | | | | | | | |
| Grok-4 | 5.7 | 2.7 | 0.0 | 0.0 | 0.0 | 3.5 | 7.7 | 3.9 | 2.9 | 0.0 | 3.3 | 5.4 |
| Claude-sonnet-4 | 7.3 | 7.2 | 8.7 | 4.7 | 15.2 | 7.7 | 14.3 | 9.5 | 14.5 | 20.4 | 27.5 | 14.7 |
| Claude-sonnet-4-thinking | 10.9 | 7.2 | 10.9 | 9.3 | 15.2 | 10.2 | 19.1 | 9.0 | 15.2 | 16.3 | 23.5 | 16.5 |
| GPT-4.1 | 12.1 | 14.4 | 13.0 | 9.3 | 30.3 | 13.8 | 21.0 | 18.9 | 24.2 | 23.7 | 43.2 | 22.6 |
| GPT-4o | 13.4 | 14.4 | 13.0 | 11.6 | 15.2 | 13.5 | 23.2 | 20.0 | 24.4 | 24.4 | 27.5 | 23.0 |
| Qwen-VL-Max | 10.5 | 13.5 | 10.9 | 16.3 | 30.3 | 13.1 | 21.4 | 19.9 | 20.3 | 3.4 | 38.4 | 23.0 |
| o4-mini | 26.3 | 23.4 | 21.7 | 18.6 | 27.3 | 24.6 | 37.3 | 39.4 | 36.5 | 35.5 | 41.7 | 35.0 |
| o3 | 27.1 | 29.7 | 15.2 | 14.0 | 36.4 | 26.0 | 37.3 | 36.1 | 26.1 | 25.2 | 44.2 | 35.4 |
| Doubao-1.5-vision-pro-32k | 27.5 | 27.9 | 19.6 | 20.9 | 27.3 | 26.2 | 41.2 | 36.7 | 30.5 | 39.5 | 42.6 | 39.1 |
| Doubao-seed-1.6-thinking | 36.8 | 27.0 | 17.4 | 39.5 | 30.3 | 32.5 | 48.4 | 33.7 | 30.9 | 49.6 | 55.8 | 43.9 |
| Gemini-2.5-flash-thinking | 42.9 | 36.9 | 21.7 | **41.9** | 48.5 | 39.8 | 54.2 | 43.1 | 36.2 | **51.6** | 64.4 | 50.4 |
| Gemini-2.5-pro-thinking | 40.1 | 41.4 | **39.1** | 39.5 | 48.5 | 40.8 | 51.3 | 48.1 | 50.0 | 49.8 | 62.6 | 51.1 |
| Doubao-seed-1.6 | 40.9 | 37.8 | 32.6 | 37.2 | 48.5 | 39.6 | 53.0 | 45.0 | 49.5 | 49.8 | 65.3 | 51.4 |
| Doubao-1.5-thinking-vision-pro | **43.3** | **43.2** | 26.1 | 32.6 | 48.5 | **41.0** | **56.2** | **52.1** | 41.0 | 49.8 | **66.7** | **53.9** |
| *Open-source MLLMs* (Image-only, CoT with 0-shot) | | | | | | | | | | | | |
| Gemma-3-4b-it | 1.2 | 1.8 | 2.2 | 0.0 | 0.0 | 1.2 | 4.2 | 2.4 | 2.9 | 0.0 | 1.0 | 3.1 |
| Gemma-3n-E4B | 2.4 | 2.7 | 4.3 | 7.0 | 6.1 | 3.3 | 8.1 | 6.6 | 11.0 | 11.0 | 15.4 | 8.8 |
| Gemma-3-27b-it | 4.5 | 4.5 | 2.2 | 2.3 | 6.1 | 4.2 | 10.0 | 6.0 | 7.6 | 9.5 | 13.1 | 9.0 |
| Kimi-VL-A3B-Instruct | 3.6 | 10.8 | 0.0 | 9.3 | 0.0 | 5.2 | 11.1 | 14.5 | 9.4 | 17.8 | 9.3 | 12.2 |
| Qwen2.5-VL-7B-Instruct | 4.0 | 9.0 | 13.0 | 4.7 | 6.1 | 6.2 | 15.0 | 14.7 | 23.2 | 18.2 | 21.7 | 16.5 |
| InternVL3-8B | 8.5 | 10.8 | 4.3 | 9.3 | 12.1 | 9.0 | 16.0 | 16.5 | 11.4 | 15.5 | 30.2 | 16.6 |
| InternVL3-14B | 7.7 | 14.4 | 8.7 | 4.7 | 21.2 | 10.0 | 15.6 | 18.7 | 20.0 | 14.3 | 35.4 | 18.0 |
| Llama-4-Maverick | 11.3 | 10.8 | 13.0 | 9.3 | 6.1 | 10.8 | 19.8 | 13.9 | 21.7 | 18.6 | 22.5 | 18.7 |
| InternVL3-78B | 7.7 | 15.3 | **15.2** | 11.6 | 15.2 | 11.0 | 17.3 | 19.1 | 24.3 | 25.6 | 34.5 | 20.3 |
| Qwen2.5-VL-32B-Instruct | 8.9 | 13.5 | 13.0 | **18.6** | 30.3 | 12.7 | 18.4 | 18.4 | 19.9 | 31.8 | 41.4 | 21.3 |
| InternVL3-38B | 10.1 | 16.2 | 8.7 | 11.6 | 24.2 | 12.5 | 19.5 | 19.7 | 15.9 | 26.2 | 42.2 | 21.4 |
| GLM-4.1v-thinking-flashx | 14.2 | 12.6 | 8.7 | 9.3 | 18.2 | 13.1 | 27.1 | 20.5 | 15.9 | 22.7 | 32.5 | 24.5 |
| Qwen2.5-VL-72B | 12.6 | **17.1** | 10.9 | 16.3 | 18.2 | 14.2 | 26.5 | 24.1 | 20.2 | **34.9** | 42.2 | 27.2 |
| ERNIE-4.5-Turbo-VL-Preview | **18.2** | 13.5 | 13.0 | 16.3 | 27.3 | **17.1** | **32.5** | 21.5 | **24.6** | 32.7 | **50.2** | **30.4** |
| *Reasoner* (Image-only, CoT with 0-shot) | | | | | | | | | | | | |
| Keye-VL-8B-Preview | 3.2 | 4.5 | 0.0 | 4.7 | 6.1 | 3.5 | 4.7 | 4.8 | 0.7 | 4.7 | 13.4 | 4.9 |
| OVR | 2.8 | 5.4 | 4.3 | 7.0 | 15.2 | 4.8 | 6.8 | 7.2 | 9.4 | 14.9 | 19.6 | 8.7 |
| Revisual-R1 | 6.1 | 6.3 | 4.3 | 4.7 | 12.1 | 6.2 | 11.9 | 7.5 | 8.7 | 9.3 | 26.0 | 11.3 |
| OpenVLThinker | 5.3 | 9.0 | 6.5 | 9.3 | 12.1 | 7.1 | 14.8 | 14.4 | 13.0 | 20.9 | 24.7 | 15.8 |
| ThinkLite-VL | 6.1 | 9.9 | 8.7 | 4.7 | 12.1 | 7.5 | 16.7 | 15.3 | 15.9 | 20.2 | 32.5 | 17.7 |
| VLAA-Thinker-Qwen2.5VL-7B | 5.7 | 10.8 | 8.7 | 7.0 | 9.1 | 7.5 | 16.0 | 17.6 | 18.2 | 22.9 | 34.0 | 18.5 |
| WeThink | 6.9 | 9.9 | **13.0** | 11.6 | 9.1 | 8.8 | 17.5 | 18.1 | **27.1** | 24.0 | 33.2 | 20.2 |
| MMR1-Math-v0-7B | 8.9 | 11.7 | 4.3 | 9.3 | 12.1 | 9.4 | 19.8 | 17.9 | 14.3 | 24.4 | 35.1 | 20.3 |
| MM-Eureka | 6.1 | **16.2** | 8.7 | 4.7 | 15.2 | 9.2 | 18.6 | 21.5 | 19.0 | 19.0 | **38.9** | 20.7 |
| MiMo-VL-7B-RL | 15.4 | 12.6 | 4.3 | 9.3 | 21.2 | 13.5 | 23.7 | 19.1 | 10.1 | 18.0 | 37.6 | 21.8 |
| VL-Rethinker-7B | 10.5 | 15.3 | **13.0** | 14.0 | 18.2 | 12.7 | 21.6 | **21.6** | 23.0 | 29.4 | 35.3 | 23.4 |
| Skywork-R1V3-38B | **18.2** | 8.1 | 8.7 | **20.9** | **21.2** | **15.4** | **30.3** | 16.1 | 18.4 | **35.3** | 34.3 | **26.6** |

representations—model-generated question text (QM) and figure description (DM)—followed by reasoning (I+QM and I+QM+DM). This framework enables systematic analysis of perception and reasoning under realistic conditions. Since in real-world scenarios users would not input models in a few-shot manner, we restrict our evaluation to the CoT with 0-shot setting only.

## 3.2 EVALUATION PROTOCOL

**Strict Accuracy ($Acc_{str}$).** $Acc_{str}$ requires that all sub-answers within a question be correct for the model to receive credit. If any sub-answer is incorrect, the entire question is marked wrong.

**Loose Accuracy (Acc).** Acc allows partial correctness and is computed as the proportion of correctly answered sub-questions within each question.

For both metrics, an automated scoring pipeline based on GPT-4.1-nano compares model answers against reference answers, enforcing strict rules for mathematical equivalence, numerical tolerance, unit consistency, and symbolic structure to ensure scalable and reliable evaluation in real-world tasks.

## 3.3 MAIN RESULTS

**Robustness Challenge Under Real-world Visual Noise.** MATHREAL presents math questions photographed in realistic settings, introducing three key types of visual degradation: image quality deterioration, viewpoint shifts, and handwritten annotations. These factors pose substantial challenges to visual understanding and reasoning for MLLMs. Evaluation reveals sharp performance disparities under these conditions. Under the Acc, the top-performing models are Doubao-1.5-thinking-vision-pro (53.9%) and Doubao-seed-1.6 (51.4%), while GPT-4o and Claude-sonnet-4 reach only 23.0% and 14.7%, respectively. At the other end of the spectrum, the weakest model, Gemma-3-4b-it, achieves just 3.1%. These results highlight the difficulty current MLLMs face in handling perceptual degradation. Performance drops are substantial even for frontier models, underscoring the limitations of current vision-language alignment and error tolerance. MATHREAL thus offers a more realistic and discriminative benchmark for evaluating robustness under imperfect, real-world inputs.

**Performance Gap Between Closed and Open Models.** Results on the MATHREAL benchmark show that closed-source models significantly outperform their open-source counterparts across all evaluation metrics and task types, with performance gaps further amplified under noisy visual inputs. Under the strict accuracy metric ($Acc_{str}$), Doubao-1.5-thinking-vision-pro achieves the highest average accuracy of 41.0%. In contrast, the best open-source model, ERNIE-4.5-Turbo-VL-Preview, reaches only 17.1%, resulting in a gap of over 20%. Reasoners also lag behind, with the strongest performer, MiMo-VL-7B-RL, reaching only 13.5% under $Acc_{str}$. Most others fall below 10%, highlighting the difficulty of integrating reasoning pipelines with robust visual perception under degraded inputs. This further emphasizes the advantage of end-to-end, well-aligned architectures in closed models when handling real-world visual challenges.

**Performance Divergence Across Categories.** MATHREAL reveals substantial performance divergences across the five categories, reflecting distinct cognitive demands and multimodal challenges. Statistical charts (SC) yield the highest accuracies under both strict and loose metrics; for example, Doubao-1.5-thinking-vision-pro achieves 48.5% $Acc_{str}$, and Doubao-seed-1.6 reaches 48.5%. These tasks benefit from structured layouts and low geometric ambiguity, enabling extraction from bar charts, tables, and plots. In contrast, logical reasoning (LR) and function graphs (FG) are the most challenging. LR involves abstract symbolic inference, with top models like Gemini-2.5-pro-thinking at 39.1% $Acc_{str}$ and Doubao-seed-1.6 at 32.6%. FG requires precise spatial alignment between visual features and expressions; even the best models, such as Gemini-2.5-flash-thinking, attain only 41.9%. Overall, models perform best when visual input is structured and symbolic reasoning is limited. Tasks requiring spatial abstraction, continuous alignment, or geometric complexity—particularly under visual noise—remain key limitations for current MLLMs.

**Model Gaps in OCR and Description Handling.** Evaluation under different input settings reveals that current models still face significant challenges in OCR and structured description understanding. While adding QM or DM brings little or even negative gains, providing QG and DG leads to substantial improvements across models. For example, Grok-4 remains below 10% accuracy with I and I+QM, yet surpasses 50% once QG or DG are provided. This clear divergence suggests that models struggle to robustly extract and structure information directly from images, but can reason effectively once accurate textual inputs are supplied. In contrast, stronger models such as Gemini-2.5-pro-thinking show incremental improvements across all settings, indicating relatively better internal perception but still benefiting from explicit QG/DG inputs. Overall, these results highlight that OCR and structured description remain bottlenecks for real-world math reasoning. Future models could address this gap by enhancing perception capabilities during pre-training, enabling post-training stages to better activate the synergy between perception and reasoning. More detailed results are provided in the Appendix.

**Real Image vs. Clean Image.** To assess model robustness to image quality, we select 175 questions from the testmini set and retrieve higher-quality clean versions of those images. We then evaluate models on both real and clean inputs, computing $\Delta = \text{Acc}_{\text{Clean}} - \text{Acc}_{\text{Real}}$ and aggregating these deltas across the fourteen interference categories with both coarse-grained (binary presence/absence) and fine-grained groupings. Most models exhibit substantial gains on clean images. Llama-4-Maverick improves by +12.0% and Claude-sonnet-4-thinking by +11.8%—indicating that visual noise significantly constrains their real-image performance. Blur attenuates the high-frequency details essential for OCR-based text extraction and fine-grained visual feature recognition, while rotation disrupts spatial alignment and forces reliance on implicit geometric transforms, causing the strict accuracy of Claude-sonnet-4-thinking and Doubao-seed-1.6 to drop by approximately –0.25 and –0.20, respectively; in contrast, models pretrained with extensive rotational augmentation, such as Gemini-2.5-pro-thinking and Qwen2.5VL-72B, remain largely unaffected. Figure marking and handwritten answer interference often highlight key regions or provide solution cues, yielding modest benefits to Doubao-1.5-thinking-vision-pro and Gemini-2.5-pro-thinking; by contrast, InternVL-3-78B and Claude-sonnet-4-thinking, which exhibit weaker visual-saliency integration, suffer slight declines. Notably, Doubao-1.5-thinking-vision-pro achieves a remarkable +0.21 increase in strict accuracy ($\text{Acc}_{\text{str}}$) on non-blurred real images versus clean versions—likely due to its vision backbone being thoroughly trained on authentic mobile-captured data, enabling it to exploit real-world lighting, shading, and texture cues.

**LLM-as-a-Judge Consistency.** To assess the reliability of automatic evaluation, we adopt the LLM-as-a-judge paradigm. Specifically, for each question we use the prompt templates detailed in the Appendix and employ the GPT-4.1-nano model to generate judgments. To validate consistency, we randomly sample 100 questions each from the main results of GPT-4o and Doubao-1.5-thinking-vision-pro under $\text{Acc}_{\text{str}}$, and manually score them according to the same prompt. A case is considered inconsistent if the manual score diverges from the GPT-4.1-nano judgment. The resulting agreement rates are 93% and 94% for GPT-4o and Doubao-1.5-thinking-vision-pro, respectively. These high consistency levels demonstrate that the LLM-as-a-judge approach provides reliable evaluation in our benchmark setting. Importantly, this method substantially reduces the labor cost of manual comparisons while maintaining robust fidelity, thus offering a scalable and cost-effective solution for both benchmark evaluation and model development.

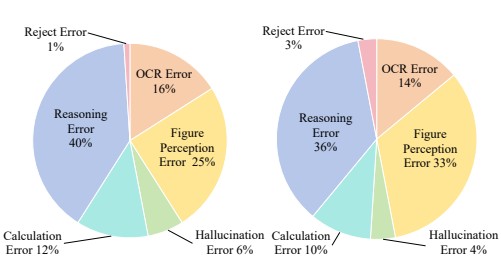

Figure 6: Error distribution over 100 annotated cases from Doubao-1.5-thinking-vision-pro (left) and Gemini-2.5-pro-thinking (right) error cases.

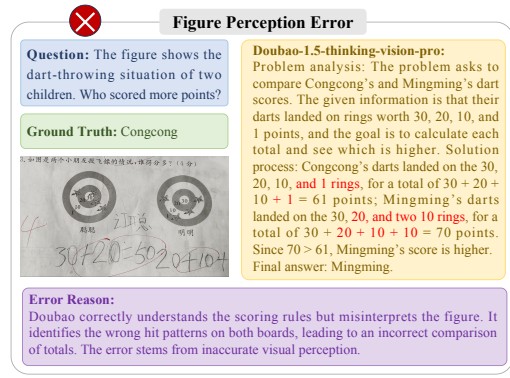

Figure 7: A basic figure perception error, with the error highlighted in red. More examples can be found in the appendix.

## 3.4 ERROR ANALYSIS

We conduct a detailed error analysis by randomly sampling 100 failed cases (Acc = 0) from each of Doubao-1.5-thinking-vision-pro and Gemini-2.5-pro-thinking. The errors are categorized into six types: OCR error, figure perception error, calculation error, reasoning error, hallucination, and reject error. The distribution is shown in Figure 6.

We observe a broadly consistent trend across both models. Reasoning errors account for the largest proportion (over one-third), indicating that even when perception is mostly accurate, models often

fail to construct valid logical chains or apply correct mathematical principles. Visual understanding remains another major source of failure. Specifically, figure perception errors and OCR errors together account for 40–50% of the failures, reflecting the strong dependence of multimodal math tasks on accurate visual decoding. In particular, noisy charts, distorted symbols, and handwritten notations frequently lead to misread digits or misinterpreted geometric structures. These perception issues are critical, as they compromise the model's input before any reasoning occurs. Calculation errors, hallucinations, and reject errors occur less frequently but still contribute to overall performance degradation. Notably, hallucinations often arise when models fabricate nonexistent quantities or assumptions, while reject errors reflect failure to produce meaningful answers under uncertainty. Overall, the findings highlight two primary challenges: robust visual understanding under imperfect inputs, and consistent multi-step reasoning over noisy or ambiguous content. Addressing either alone is insufficient—future progress in MLLMs will require tightly integrated improvements across perception, parsing, and reasoning components.

## 4 RELATED WORK

**Plain Text Benchmarks.** MathQA Amini et al. (2019) is a large-scale benchmark consisting of math word problems designed to evaluate problem-solving in arithmetic and algebra through natural language. GSM8K Cobbe et al. (2021) contains 8,500 elementary-level math problems that test multi-step reasoning. In contrast, MATH Hendrycks et al. (2021) provides 12,500 challenging high-school competition-level questions. SuperCLUE-Math Xu et al. (2024) specializes in Chinese mathematical reasoning tasks. RV-Bench Hong et al. (2025) evaluates structural understanding by programmatically replacing numerical values in problems. Math-RoB Yu et al. (2025) introduces controlled perturbations to assess model stability under variations. PolyMath Wang et al. (2025e) addresses this by providing a high-quality, large-scale multilingual evaluation set.

**Multimodal Benchmarks.** With the development of multimodal large models, many benchmarks focused on multimodal math problems have also emerged. MathVista Lu et al. (2023) establishes the first comprehensive multimodal math evaluation through 6,141 visual tasks across diverse mathematical reasoning scenarios. MathVerse Zhang et al. (2024b) advances visual understanding assessment through 15,000 diagram-based samples, specifically designed to quantify diagram utilization in math problem-solving. MATH-Vision Wang et al. (2024) elevates evaluation standards with 3,040 competition-grade problems, creating a rigorous testbed for advanced mathematical reasoning. VisOnlyQA Kamoi et al. (2024) reveals fundamental limitations in geometric perception through 12 tasks demonstrating that even SOTA models struggle with basic visual perception. MathGlance Sun et al. (2025) isolates mathematical perception evaluation through 1,200 images and 1,600 questions spanning core perceptual tasks. MV-MATH Wang et al. (2025c) challenges the multivisual reasoning by developing 2,009 multi-image problems mirroring real-world mathematical contexts. GeoEval Zhang et al. (2024a) emphasizes unseen dataset evaluation importance through 2,000 geometry problems with specialized subsets for comprehensive assessment. We-Math Qiao et al. (2024) introduces four-dimensional evaluation metrics for knowledge acquisition and generalization assessment through 6,500 visual problems spanning 67 hierarchical concepts. CMMath Li et al. (2024b) delivers the first native Chinese mathematical benchmark with 23,000 curriculum-aligned questions, filling the critical gap in K-12 educational assessment.

## 5 CONCLUSION

MATHREAL introduces a new benchmark for evaluating MLLMs on real-world, noisy images of K–12 math questions, addressing the limitations of existing benchmarks that rely on clean images. The dataset includes diverse math questions with various types of visual noise, such as blur, perspective distortions, and handwritten interference. By evaluating several open-source and closed-source models, we establish a benchmark that highlights the limitations of current MLLMs in multi-visual mathematical reasoning, emphasizing the impact of image quality, input methods, and question types on performance. Our analysis reveals that most models struggle with noisy images, pointing to the need for more robust visual encoders in MLLMs. This work sets the stage for future improvements in multimodal reasoning, especially in real-world educational settings.

ETHICS STATEMENT

This research does not involve human subjects, personal data, or sensitive information. The MATH-REAL benchmark is built from photographs of educational materials and anonymized question repositories, with careful filtering to exclude any potentially identifying or private content. All images depict only mathematical questions and related figures, and no faces, personal information beyond problem-solving steps, or metadata are retained. The dataset is intended strictly for academic research and evaluation of multimodal reasoning systems, and we believe it poses no foreseeable ethical risks.

REPRODUCIBILITY STATEMENT

We have taken multiple steps to support reproducibility of our results. The dataset construction pipeline, including collection, filtering, and multi-stage manual annotation, is documented in the main paper and appendix. We provide taxonomy definitions, evaluation metrics, and scoring scripts, together with configuration details for all experimental settings. The *testmini* split and full annotation metadata are released at submission to allow method development and ablation studies. The complete dataset will be made publicly available upon acceptance. We also release prompts and evaluation templates to facilitate exact replication wherever model APIs allow.

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

## Supplementary Material

### Supplementary Material Overview

- Section A: Introduction.
- Section B: Related work.
- Section C: Dataset Details.
- Section D: Experimental Details.
- Section E: Results Analysis.

## A   Introduction

The data and code: `https://anonymous.4open.science/r/MathReal-52CD`

## B   Related Work

### B.1   Benchmark for Perception and OCR as the Foundation of Reasoning

DocVQA Mathew et al. (2021) introduces 28,000 real document QA pairs, establishing the first visual question answering evaluation framework for structured documents like contracts and reports. ChartQA Masry et al. (2022) develops 3,200 chart QA samples, pioneering the joint reasoning evaluation mechanism between axis text and visual elements. SEED-Bench-2-Plus Li et al. (2024a) expands to 15,672 test samples covering three rich-text environments, enabling fine-grained evaluation across 63 data types. Fox Liu et al. (2024b) introduces 9 specialized sub-tasks including region-level OCR and color-guided text recognition, establishing the first benchmark for fine-grained document understanding across multi-page layouts. MMTab Zheng et al. (2024) releases 5,000+ tax/medical form test sets with specialized metrics for complex table reasoning like merged cells and cross-column references. CC-OCR Yang et al. (2024) collects 15,000 cross-language text

images, supporting complex document parsing validation across LaTeX, HTML and SMILES formats. OCR-Reasoning Huang et al. (2025) creates 1,069 advanced reasoning questions with only 2.3% directly extractable answers, specifically testing deep reasoning capabilities like spatial relationships and numerical calculations. OCRBench v2 Fu et al. (2024) upgrades to 10,000 human-verified QA pairs across 31 scenarios and 23 tasks, first integrating eight core capability assessments including text localization and logical reasoning.

## C  DATASET DETAILS

### C.1  DATA ANNOTATION PROCESS

To facilitate annotation, we develop a Gradio-based data annotation platform and organize the process into three fully manual stages: e-screening of basic image content, annotation of image conditions, annotation of question-level metadata. This structured workflow ensures high semantic and structural quality while reflecting the complexity and diversity of real-world educational scenarios.

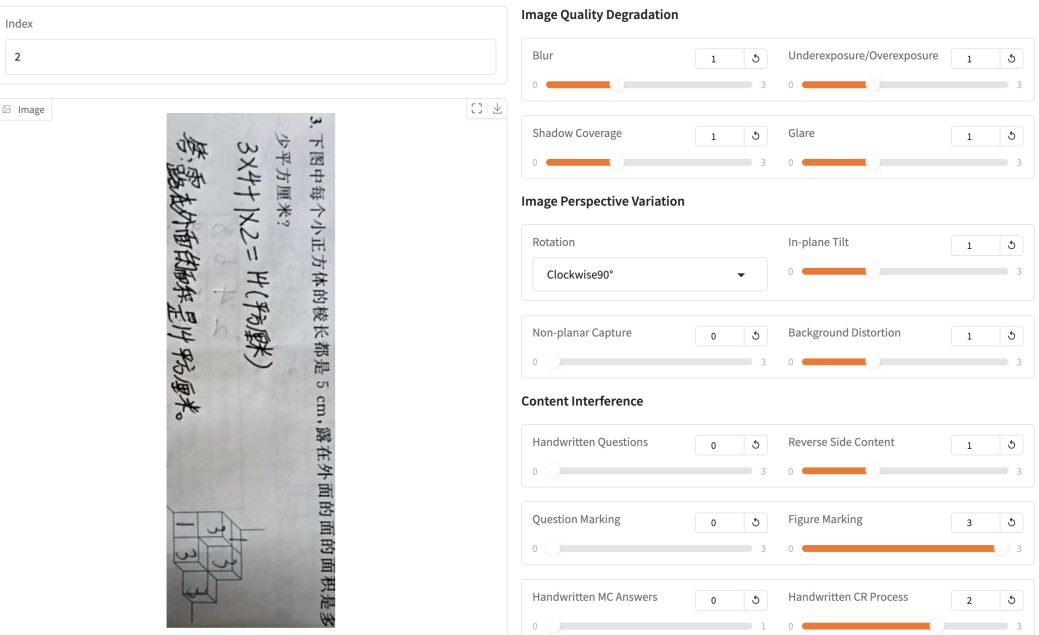

Figure 8: Gradio annotation page of stage two.

**Stage One – Re-screening.** We manually verify whether each sample satisfies the three conditions established during data collection:

- *Single Question Only*: the image contains exactly one complete question, with possible interference from other incomplete or partial questions.
- *Complete Question*: the question text and figure are fully visible, with no missing text or critical contents.
- *Figure Relevant to Solution*: the diagram or figure is essential for understanding or solving the problem, not merely decorative or incidental.

Samples that fail to meet any of these criteria are discarded. This step ensures that only valid, solvable, and diagram-dependent math questions proceed to the next stage.

**Stage Two – Real-world scenario categories and levels.** We annotate each image according to a fine-grained taxonomy of real-world scenario categories and levels. This taxonomy comprises three primary categories with fourteen subcategories:

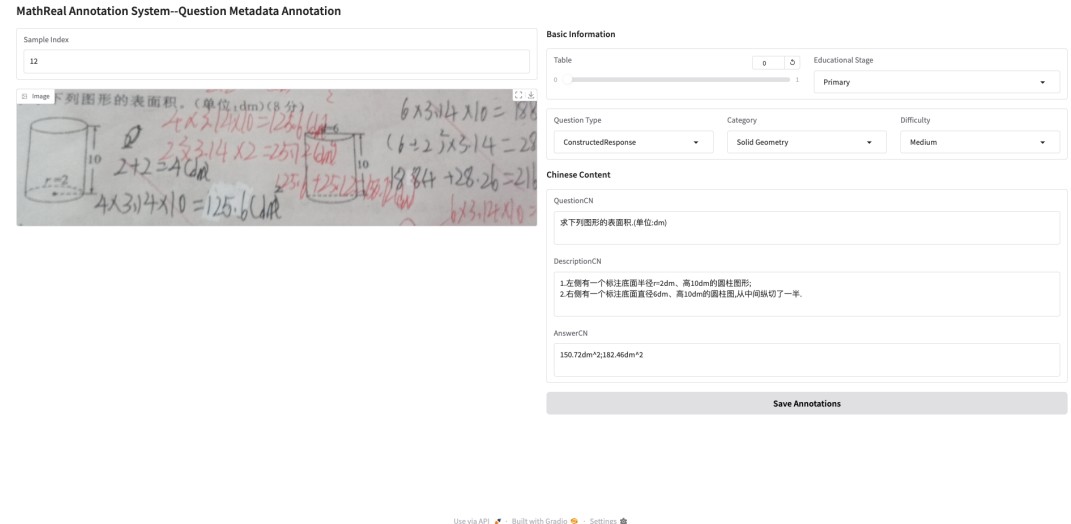

Figure 9: Gradio annotation page of stage three.

- Image Quality Degradation:
  - *Blur* (0–3): The degree to which the image's text and figures are visually out of focus, ranging from completely clear and legible to entirely unrecognizable. 0: completely sharp and all text clearly legible, 1: slight blur but content recognizable, 2: strong blur making recognition difficult, 3: severe blur rendering content unreadable.
  - *Underexposure/Overexposure* (0–3): The extent of excessive darkness or brightness in the image that may obscure content, from no exposure issues to fully black or white images. 0: no brightness issues, 1: mild darkness or brightness with content still visible, 2: severe underexposure or overexposure partially obscuring content, 3: extreme exposure resulting in completely black or white image.
  - *Shadow Coverage* (0–3): The proportion of the question area obscured by shadows, from none to more than 60% coverage. 0: no shadows, 1: shadows covering 1%–30% of the content, 2: shadows covering 30%–60%, 3: shadows covering more than 60%.
  - *Glare* (0–3): The presence of reflected light spots on the image, ranging from none to severe glare that renders the content unreadable. 0: no glare, 1: minor glare with text still legible, 2: strong glare partially obscuring content, 3: severe glare rendering content unreadable.
- Image Perspective Variation:
  - *Rotation*: The orientation of the image compared to a correctly aligned version. (Upright, clockwise $90°$, counterclockwise $90°$, or $180°$)
  - *In-plane Tilt* (0–3): The tilt angle of the image within the xy-plane, from no tilt to a tilt angle greater than $30°$. 0: no tilt, 1: tilt angle within $15°$, 2: tilt angle between $15°–30°$, 3: tilt angle greater than $30°$.
  - *Non-planar Capture* (0–3): Perspective distortion caused by capturing the image from a non-perpendicular angle, resulting in trapezoidal or irregular shapes. 0: no perspective distortion, 1: slight perspective distortion without recognition difficulty, 2: trapezoidal or irregular deformation with partial recognition impact, 3: severe deformation such as ladder-shaped or warped forms strongly affecting recognition.
  - *Background Distortion* (0–3): Physical bending or warping of the background or paper, from flat to severely deformed shapes affecting content recognition. 0: flat background, 1: minor folding without recognition impact, 2: moderate warping causing partial deformation, 3: severe bending or curling with strong recognition interference.
- Irrelevant Content Interference:
  - *Handwritten Questions* (0–3): The extent to which the question text is handwritten, from neatly written to extremely illegible. 0: printed text, 1: neatly handwritten text,

2: irregular handwriting with recognition difficulty, 3: extremely messy handwriting almost illegible.

- *Reverse-side Content* (0–3): Visual interference from text or images on the reverse side of the paper, from none to severe bleed-through. 0: no interference, 1: slight bleed-through without impact, 2: large amount of bleed-through partially obscuring content, 3: severe bleed-through completely obscuring front content.

- *Question Marking* (0–3): The presence of underlining, circling, or other markings on the question text, from none to heavily marked. 0: no markings, 1: few markings with minimal interference, 2: frequent markings moderately obscuring text, 3: heavy markings over most of the text.

- *Figure Marking* (0–3): Markings drawn on figures, from none to extensive markings obscuring geometric shapes. 0: no markings, 1: one marked element not affecting recognition, 2: multiple markings partially obscuring shapes, 3: extensive markings heavily obscuring geometric figures.

- *Handwritten Answers for Multiple-choice or Fill-in-the-blank Questions* (0–1): The presence of handwritten answers in answer blanks or options. 0: no handwritten answers, 1: presence of handwritten answers.

- *Handwritten Process for Constructed-response Questions* (0–3): The amount of handwritten solution steps shown in the image, from none to four or more lines. 0: no solution steps, 1: one line of steps, 2: two to three lines of steps, 3: four or more lines of steps.

We provide detailed annotations for each subtype to support fine-grained analysis of model robustness under diverse real-world conditions.The gradio page of this stage is in Figure 8.

**Stage Three – Question Metadata Annotation.** We annotate eight key attributes:

- Ground-truth Question: The printed question text exactly as it appears in the image.

- Presence of Tables: Whether the question contains any tabular data (0 for no, 1 for yes).

- Educational Level: The intended education stage, categorized as primary, middle, or high school.

- Question Type: The answer format, including multiple-choice, fill-in-the-blank, or constructed-response.

- Category: The primary domain of the question, including plane geometry (PG), solid geometry (SG), logical reasoning (LR), function graphs (FG), and statistical charts (SC).

- Ground-truth Answer: The correct answer verified by annotators.

- Figure Description: A detailed natural-language description of the figure, excluding any question text.

- Clean Image: A standardized and clean version of the image retrieved via web search when available.

The gradio page of this stage is in Figure 9.

Finally, we conduct a fully human-verified review to ensure consistency and accuracy across all stages. Through this three-stage pipeline, we construct MATHREAL, a high-quality dataset of real-world, diagram-based math questions that provides a rigorous benchmark for evaluating visual perception and reasoning under authentic conditions.

### C.2 QUESTION DISTRIBUTION

All questions in the dataset are presented in Chinese. The longest question contains 451 characters, while the shortest has only 7 characters, with an average length of 122.03 characters. Figure 10 further illustrates the distribution of question lengths, revealing a diverse range from very short prompts to extended, detailed questions.

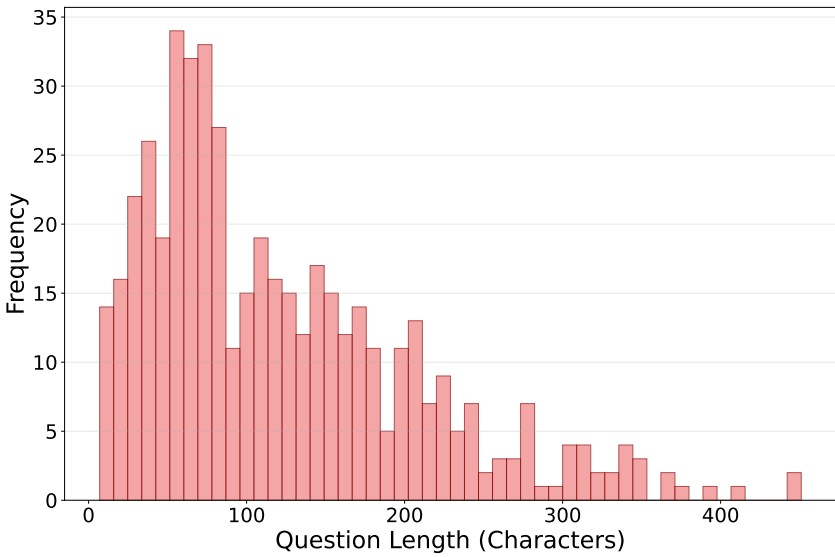

Figure 10: QuestionCN Length Distribution.

# D  EXPERIMENTAL DETAILS

## D.1  PROPMT FOR OCR AND FIGURE UNDERSTANDING GENERATION

This prompt is designed to separately guide multimodal large language models in performing OCR-based question text extraction and detailed figure understanding for real-world, image-based mathematical problems. The OCR Task section specifies strict recognition rules, focusing solely on printed question stems while excluding handwritten content, metadata, and irrelevant figure text. It enforces format preservation, standardized handling of blanks, and precise processing of tables, ensuring faithful reproduction of textual content without interpretation or solution attempts. The Figure Understanding Task section instructs the model to analyze only the mathematical figures—such as geometric diagrams, function plots, and statistical charts—present in the image. It requires a comprehensive, standalone description that details the figure's structure, key elements, and mathematical properties, without solving the problem or performing OCR. Together, these prompts enable a clear separation between textual content extraction and visual element analysis, supporting controlled evaluations of perception and reasoning.

## D.2  PROMPT FOR ANSWER GENERATION

In our study, we design six experimental settings (**I**, **I$_{UER}$**, **I+QM**, **I+QM+DM**, **I+QG**, and **I+QG+DG**) to progressively disentangle visual perception and reasoning, enabling a systematic evaluation of MLLMs' perception and reasoning abilities under realistic educational scenarios. To operationalize these settings and ensure consistency across experiments, we develop task-specific prompts that guide the models in processing visual and textual information in a controlled manner.

The **Main Setting Prompt** is used for the primary evaluation setting (**I**), where the model receives only the raw image and is required to jointly perform visual perception and mathematical reasoning. The instructions are structured to guide the model from problem analysis, through detailed reasoning, to a strictly formatted final answer, ensuring that all information in the image is effectively utilized.

The **I$_{UER}$ Setting Prompt** is tailored for the unified end-to-end reasoning scenario, where the model performs OCR, figure understanding, and solution derivation within a single interaction. The workflow in this prompt is explicitly divided into OCR extraction, detailed figure analysis, reasoning, and final answer formatting. By combining perception and reasoning within a unified instruction set, this prompt facilitates systematic assessment of a model's ability to integrate multimodal information in a one-pass pipeline under real-world conditions.

Table 3: Prompt for OCR Task and Figure Understanding Task.

| **Prompt for OCR Task** |
| --- |

You are a professional OCR text recognition expert. Please strictly follow the instructions below:
1. Recognition Scope:
    Recognize only the printed question stem in the image. Ignore any handwritten content. Include only the question stem, excluding the problem number, year, region, and score.
2. Output Format:
    Output text according to the original layout in the image, preserving paragraphs and line breaks. Do not merge or split paragraphs arbitrarily.
3. Multiple-Choice Options:
    – If the option content consists only of text or numbers, fully recognize and output the options and their corresponding content.
    – If any option contains image elements, do not recognize or output any option content.
4. Fill-in-the-Blank Questions:
    – If blanks are present, represent them uniformly as "____" (four underscores).
    – If blanks are parentheses that need to be filled, represent them uniformly as "(    )" (two parentheses and four spaces).
5. Math Questions with Figures:
    – If text in the figure consists only of numbers, letters, or labels (e.g., AB, 30°), do not recognize or output it.
    – Ignore all text embedded in abstract graphics (e.g., geometric figures, statistical charts, function plots); do not include it in the question stem.
6. Figure Captions:
    Ignore all figure captions; do not recognize or output them.
7. Table Processing:
    – Recognize text in the table row-by-row according to its original order.
    – Use a single space as the delimiter between columns (e.g., "No. Name 1 Zhang San 2 Li Si").
Important Notes!!
    – Only return the actual recognized text content.
    – Do not add any explanations, analysis, hints, or extra notes.
    – Do not solve the problem or return the answer.
    – No image analysis is required; directly return the OCR results only.

| **Prompt for Figure Understanding Task** |
| --- |

You are a professional mathematical figure analysis expert. Please analyze the mathematical figure in the image and provide a detailed description.
Requirements:
    1. Analyze only the mathematical figures in the image, including geometric figures, function plots, and statistical charts.
    2. Describe in detail the basic features, key elements, and mathematical properties of the figure.
    3. Your answer should contain only one part: *description*.
    4. The description must clearly and thoroughly describe the elements, structures, geometric shapes, or chart contents in the figure.
    5. Do not solve the problem or perform OCR recognition; only analyze what is present in the figure itself.
Directly output the *description* without adding any extra content, explanations, or hints.

Table 4: Prompt for Response Generation.

| **Main Setting Prompt for Response Generation** |
| --- |

Please solve the problem in the image by following these steps, and do not refuse to answer:
   1. Problem Analysis: Clearly identify the problem requirements, known conditions, and the objective to be solved from the image.
   2. Solution Process:
      (1) Fully utilize the information provided in the image.
      (2) Present the reasoning and calculation process in detail.
      (3) Explain the principles behind each key step.
      (4) Perform verification or validation when necessary.
   3. Final Answer:
      (1) Place the answer inside \boxed{}.
      (2) If there are multiple answers, place each one inside a separate \boxed{}.
      (3) Strictly follow the required format for numerical values, units, etc., as stated in the problem.

| **I$_{\text{UER}}$ Setting Prompt for Response Generation** |
| --- |

Please answer the following math problem and strictly follow the steps below. Do not refuse to answer.
1. OCR of the Question Text:
   Scope
      – Recognize only the printed question stem in the image.
      – Ignore any handwritten content.
      – Exclude problem number, year, region, and score.
   Output Format
      – Preserve the original layout, paragraphs, and line breaks.
      – Do not merge or split paragraphs arbitrarily.
      – Use English punctuation only.
   Multiple-Choice Questions
      – If options are text or numbers, recognize and output them completely.
      – If options contain image elements, do not output any options.
   Fill-in-the-Blank Questions
      – Represent blanks with "____" (four underscores).
      – Represent to-be-filled parentheses with "(    )" (two parentheses with four spaces).
   Questions with Figures
      – Ignore pure digits, letters, and labels inside the figure.
      – Do not OCR text embedded in abstract graphics such as geometric figures, statistical charts, or function plots.
   Special Handling
      – Figure captions: ignore completely.
      – Dialogue-style context images: recognize only the question stem and ignore dialogues in the image.
      – Tables: recognize row by row in the original order; separate columns with a single space.
   Notes
      – Recognize text content only.
      – Do not add any explanations, analyses, or hints.
2. Figure Understanding:
      – Analyze only the mathematical graphics in the image, including geometric figures, function plots, and statistical charts.
      – Describe the basic characteristics, key elements, and mathematical properties of the figure in detail.
      – Your output should contain a single section named *description*.
      – *Description* must detail the elements, structures, geometric shapes, or chart content present in the figure.
      – Do not solve the problem and do not perform OCR here; only analyze the figure content.
3. Solution Process:
      (1) Fully utilize information from the image, the OCR step, and the figure understanding step.
      (2) Present the reasoning and calculation steps in detail.
      (3) Explain the principles behind each key step.
      (4) Perform verification or validation when necessary.
4. Final Answer:
      (1) Place the answer inside \boxed{}.
      (2) If there are multiple answers, place each one inside a separate \boxed{}.
      (3) Strictly follow the required format for numbers, units, and other specifications as stated in the problem.

Table 5: Prompt for Answer Extraction Task.

---

**Prompt for Answer Extraction Task**

---

You are a professional answer extraction expert. Please extract the final answer from the following text as accurately as possible, strictly following the priority strategy below:

Priority 1: Look for explicit answer keywords
- Search for the following keywords:
    * "final answer", "answer", "result"
    * "the answer is", "the result is"
    * Summary words such as "therefore", "so", "in conclusion" followed by the answer content
    - Extract the content that immediately follows these keywords
Priority 2: Extract from the end of the text
    - If no explicit answer is found in the previous step, try to extract the most likely answer from the last paragraph or last sentence of the text
Important Requirements:
    1. Multiple answers should be separated by semicolons (;)
    2. Return only the answer content itself, without extra explanations or formatting
    3. If the answer cannot be determined, return *null*
Strictly follow the above priority order for extraction.

---

### D.3 PROMPT FOR EXTRACT AND EVALUATE ANSWERS

To ensure consistent and objective measurement of model performance across all six experimental settings, we design a two-stage evaluation pipeline comprising an *Answer Extraction* step followed by an *Answer Evaluation* step.

In the extraction stage, we first apply direct string matching to capture any content enclosed in \boxed{} from the model output. If no such match is found, we invoke a dedicated answer extraction prompt to identify the final answer based on explicit keyword matching or, failing that, from the concluding part of the output.

In the evaluation stage, the extracted answer is compared against the reference answer using a mathematical answer evaluation prompt, which enforces strict equivalence rules on numerical values, algebraic expressions, units, and multiple-part answers, while supporting proportional partial credit for partially correct responses. This design enables scalable, fine-grained, and reproducible accuracy assessment under realistic educational conditions.

### D.4 EVALUATION PROTOCOL

**OCR Accuracy Evaluation.** In real-world multimodal settings, OCR quality is often compromised by noise, handwriting, or layout distortions. To assess the reliability of model-generated OCR outputs, we adopt a hybrid metric that combines five components: numeric accuracy, keyword accuracy, semantic similarity, format and structure accuracy, and a lexical term based on normalized Levenshtein distance.

The final score is computed as:

$$\text{Acc}_{\text{OCR}} = 0.2 \cdot \text{Acc}_{\text{num}} + 0.2 \cdot \text{Acc}_{\text{keyword}} + 0.2 \cdot \text{Sim}_{\text{sem}}$$
$$+ 0.2 \cdot \text{Acc}_{\text{format}} + 0.2 \cdot (1 - \text{Lev}_{\text{norm}})$$

Here, $\text{Acc}_{\text{num}}$ measures exact agreement on all numbers and units, $\text{Acc}_{\text{keyword}}$ evaluates proper nouns and other key entities, $\text{Sim}_{\text{sem}}$ reflects sentence-level meaning consistency, and $\text{Acc}_{\text{format}}$ assesses structural fidelity (tables, paragraphs, lists). $\text{Lev}_{\text{norm}}$ is the normalized Levenshtein distance between the OCR output and the ground-truth question text. The first four scores are in $[0, 1]$ following the rubric above (with semantic decisions based on GPT-4.1-nano judgments), and the lexical component contributes via $(1 - \text{Lev}_{\text{norm}})$.

Table 6: Prompt for Mathematical Answer Evaluation Task.

---

**Prompt for Mathematical Answer Evaluation Task**

You are a top-tier mathematics evaluation expert, tasked with rigorously and precisely determining the correctness of model-generated answers.

**Core Task**

Determine whether the "Model Answer" below is mathematically and option-wise completely equivalent to the "Reference Answer", and assign a **partial credit score** based on the proportion of correct components.

**Evaluation Principles**

1. **Numerical Core Priority**:
   - Focus solely on the final numerical values, expressions, options, or conclusions.
   - Ignore solution processes, explanatory text (e.g., "the answer is:", "therefore the result is:"), variable names (e.g., D, E, Q1), and irrelevant descriptions.
   - Only retain mathematical content that directly corresponds to the reference answer for comparison.

2. **Mathematical Equivalence (Strict Judgment)**:
   - Fractions and decimals: 1/2 is equivalent to 0.5; 1/2 is equivalent to 5/10.
   - Numerical formats: 10 is equivalent to 10.0; 1,887,800 is equivalent to 1887800 (ignore thousand separators).
   - Special symbols: $\pi$ is equivalent to 3.14 (only when the problem explicitly allows approximation).
   - Algebraic expressions: $x^2+y$ is equivalent to $y+x^2$; however, `18+6\sqrt{3}` and `18-6\sqrt{3}` are **not equivalent**.
   - Formatting: $(\sqrt{3} + 3)/2$ is equivalent to $\sqrt{3}/2 + 3/2$.
   - Range notation: $x \in [0, 1]$ is equivalent to $0 \le x \le 1$.
   - **Operator Sensitivity**: $+$, $-$, $\times$, $\div$, $\wedge$ (power), etc., must be strictly consistent; any symbol error renders the expressions non-equivalent.
   - **Coordinate Points**: $(x, y)$ values must be numerically identical. Treat $x$ and $y$ as **two sub-components**. If one is correct and the other wrong, assign 0.5 for that point.
   - **Whitespace-induced formatting differences**: "y=2x+3" and "y = 2 x + 3" are equivalent; ignore the impact of spaces within expressions.

3. **Unit Handling**:
   - Reference answer has no unit: if the model answer includes a correct and reasonable unit (e.g., 15 vs 15m), it is considered correct.
   - Reference answer has a unit: incorrect units are considered wrong (e.g., 15m vs 15cm); if the model answer lacks a unit but the numerical value is correct, it is considered correct.
   - Ignore unit formatting differences: "180 { dm}$^2$" and "180dm$^2$" are equivalent; correctly extract the content.

4. **Handling Multi-Part Answers (Critical!)**:
   - You must **split the reference answer into all sub-answers (blanks)** based on its structure.
   - Each newline "\n", semicolon ";", or major section "(1)", "(2)" indicates a separate blank.
   - For each blank, further decompose it if it contains multiple components:
     • **"Or"-connected answers**: e.g., "5 or -75" → two valid solutions. If model answers only "5", give 0.5 for that blank.
     • **Coordinate pairs**: e.g., $(5, 0)$ → treat as two values. If model says $(5, 1)$, give 0.5.
     • **Multiple points**: e.g., $(1, 0), (9, 8), (-1, 9)$ → three points. Each correct point gives $1/3$.
   - Total score = sum of all correct sub-components / total number of sub-components.
   - Always allow **proportional partial credit** unless explicitly stated otherwise.

5. **Special Rules for Multiple-Choice Questions**:
   - If the reference answer is a single option (e.g., "B"), then as long as the model answer contains that option letter (e.g., "B", "B.", "Option B", "B. $f'(x_0) > g'(x_0)$") and no other options, it is considered correct → 1.0.
   - If multiple options appear or an incorrect option is selected, it is considered wrong → 0.0.

6. **Semantic Equivalence**:
   - Even if the phrasing differs, as long as the mathematical meaning is the same, it is considered correct.

7. **Proof or Graphing Questions**:
   - If the question type is a proof or graphing question, treat the model answer as acceptable by default; do not score it, and directly return `<score>1.0</score>`.

**Scoring Criteria**

   - **1.0**: All components are correct.
   - **0.0–1.0**: Assign partial credit **proportionally** based on the number of correct sub-components.
   - **0.0**: No component is correct.
   - Round to **two decimal places** (e.g., 0.83, 0.67, 0.50).

**Output Format**

You must strictly return only the XML tag containing the score, with no additional text or explanation.
```
<score>score</score>
```

---

**Answer Accuracy Evaluation.** $\text{Acc}_{\text{str}}$ requires that all sub-answers within a question be correct for the model to receive credit. If any component is incorrect, the entire question is marked as wrong. This metric emphasizes the completeness and consistency of chain-of-thought reasoning and aligns with the standard pedagogical principle of "full marks only if fully correct." It is formally defined as:

$$\text{Acc}_{\text{str}} = \frac{1}{N} \sum_{i=1}^{N} \mathbb{I} \left[ \forall j \in \{1, \ldots, K_i\}, \ a_{i,j}^{\text{pred}} \equiv a_{i,j}^{\text{gt}} \right]$$

Here, $N$ denotes the total number of questions, $K_i$ is the number of answer blanks in the $i$-th question, $a_{i,j}^{\text{pred}}$ and $a_{i,j}^{\text{gt}}$ denote the model-predicted and ground truth answers for the $j$-th blank, respectively. The indicator function $\mathbb{I}[\cdot]$ returns 1 if the condition is satisfied, and $\equiv$ denotes mathematical equivalence.

**Acc** permits partial correctness and is calculated based on the proportion of correctly predicted sub-answers within each question. This metric captures the model's partial understanding and reasoning ability under imperfect outputs:

$$\text{Acc} = \frac{1}{N} \sum_{i=1}^{N} \left( \frac{1}{K_i} \sum_{j=1}^{K_i} \mathbb{I} \left[ a_{i,j}^{\text{pred}} \equiv a_{i,j}^{\text{gt}} \right] \right)$$

### D.5 EVALUATION MODELS

We evaluate the performance of a diverse set of models on the MathReal benchmark, categorized into four groups: (a) *Large Language Models (LLMs)*, serving as text-only baselines, including Deepseek-v3 Liu et al. (2024a), Deepseek-r1 Guo et al. (2025), Qwen3 Yang et al. (2025a) and Qwen3-thinkingYang et al. (2025a); (b) *Closed-source Multimodal Large Language Models (MLLMs)*, including Grok-4 xAI (2025), Claude-sonnet-4 Anthropic (2025), Claude-sonnet-4-thinking Anthropic (2025), GPT-4.1 OpenAI (2025a), GPT-4o OpenAI (2024), o3 OpenAI (2025b), o4-mini OpenAI (2025b), Qwen-VL-MaxBai et al. (2023), Gemini-2.5-flash-thinkingComanici et al. (2025), Gemini-2.5-pro-thinkingComanici et al. (2025), Doubao-1.5-vision-pro ByteDance (2025b), Doubao-1.5-thinking-vision-pro ByteDance (2025a), Doubao-seed-1.6 ByteDance (2025c), Doubao-seed-1.6-thinking ByteDance (2025d); (c) *Open-source MLLMs*, including Gemma-3-4b-it Team et al. (2025b), Gemma-3-27b-it Team et al. (2025b), Gemma-3n-e4b Team et al. (2025b), Qwen2.5VL-7BBai et al. (2025), Qwen2.5VL-32BBai et al. (2025), Qwen2.5VL-72BBai et al. (2025), InternVL-3-8BZhu et al. (2025), InternVL-3-14BZhu et al. (2025), InternVL-3-38BZhu et al. (2025), InternVL-3-78BZhu et al. (2025), Kimi-VL-A3B-InstructTeam et al. (2025c), Llama-4-Maverick AI (2025a), GLM-4.1v-thinking-flashx AI (2025b), and ERNIE-4.5-VL-28B-A3B-PT Baidu (2025); and (d) *Multimodal Reasoning Models*, including Keye-VL Team et al. (2025d), OVR Wei et al. (2025), Revisual-R1 Chen et al. (2025b), Skywork-R1V3 Shen et al. (2025), OpenVLThinker Deng et al. (2025), ThinkLite-VL Wang et al. (2025d), VLAA-Thinker Chen et al. (2025a), WeThink Yang et al. (2025b), MMR1-Math-v0 Leng et al. (2025), MM-Eureka Meng et al. (2025), MiMo-VL-7B-RL Team et al. (2025a), and VL-Rethinker Wang et al. (2025a).

## E   RESULTS ANALYSIS

### E.1   RESULTS BY QUESTION TYPES

Table 9–11 compare model performances across three question types using the loose accuracy (Acc) average (Avg) as the primary metric. The analysis here focuses on multimodal closed-source, open-source, and reasoning-oriented models.

**Multiple-choice.** Overall accuracy is relatively low, with the best-performing model Doubao-seed-1.6 achieving an Avg of 42.3. The second-best closed-source model, Gemini-2.5-pro-thinking,

Table 7: The Acc of the OCR and the six experimental settings of models.

| Model | Acc$_{OCR}$ | I | I$_{UER}$ | I+QM | I+QG | I+QM+DM | I+QG+DG |
|---|---|---|---|---|---|---|---|
| GLM-4.1v-thinking-flashx | 81.8 | 24.5 | 19.6 | 24.9 | 32.5 | 22.1 | 34.9 |
| Qwen-VL-Max | 87.0 | 23.0 | 23.0 | 26.0 | 28.1 | 24.8 | 35.1 |
| ERNIE-4.5-turbo-vl | 89.8 | 30.4 | 30.5 | 28.4 | 32.7 | 27.8 | 36.6 |
| Llama-4-Maverick | 71.0 | 18.7 | 20.5 | 18.8 | 32.2 | 18.0 | 38.2 |
| GPT-4o | 78.6 | 23.0 | 22.4 | 22.7 | 32.2 | 24.5 | 38.7 |
| GPT-4.1 | 79.2 | 22.6 | 22.9 | 21.5 | 37.7 | 19.1 | 40.8 |
| Claude-sonnet-4 | 54.0 | 14.7 | 13.8 | 15.0 | 36.5 | 15.2 | 45.1 |
| Claude-sonnet-4-thinking | 53.9 | 16.5 | 13.7 | 15.6 | 40.5 | 13.5 | 46.9 |
| Doubao-1.5-vision-pro | 87.8 | 39.1 | 39.2 | 35.8 | 44.1 | 36.7 | 51.8 |
| o4-mini | 81.9 | 35.0 | 24.4 | 34.5 | 48.6 | 30.9 | 55.8 |
| Grok-4 | 35.6 | 5.4 | 7.7 | 9.7 | 45.8 | 9.3 | 57.7 |
| Gemini-2.5-flash-thinking | 89.8 | 50.4 | 51.5 | 51.4 | 54.0 | 49.2 | 58.3 |
| o3 | 78.4 | 35.4 | 32.0 | 33.0 | 47.8 | 34.2 | 58.5 |
| Doubao-seed-1.6-thinking | 87.9 | 43.9 | 46.2 | 45.8 | 59.5 | 46.9 | 63.2 |
| Doubao-1.5-thinking-vision-pro | 89.8 | 53.9 | 56.9 | 52.6 | 61.7 | 53.3 | 64.1 |
| Doubao-seed-1.6 | 89.7 | 51.4 | 43.8 | 52.5 | 59.5 | 48.3 | 64.2 |
| Gemini-2.5-pro-thinking | 94.0 | 51.1 | 57.4 | 59.3 | 62.0 | 61.9 | 66.0 |

Table 8: Acc Comparison: Clean vs. Real, where $\Delta = \text{Acc}_{\text{Clean}} - \text{Acc}_{\text{Real}}$.

| Model | Real | Clean | $\Delta$ |
|---|---|---|---|
| Grok-4 | 5.6 | 12.7 | +7.1 |
| Qwen2.5VL-7b | 18.2 | 20.0 | +1.8 |
| InternVL3-14b | 21.2 | 21.8 | +0.6 |
| InternVL3-8b | 18.6 | 23.3 | +4.7 |
| InternVL3-38b | 20.6 | 25.1 | +4.5 |
| Claude-sonnet-4 | 15.8 | 26.7 | +10.9 |
| InternVL3-78b | 23.1 | 29.0 | +5.9 |
| GPT-4.1 | 22.9 | 29.7 | +6.8 |
| GPT-4o | 24.1 | 31.0 | +6.9 |
| Claude-sonnet-4-thinking | 20.1 | 31.8 | +11.7 |
| Llama-4-Maverick | 18.5 | 31.8 | **+13.3** |
| Qwen-VL-Max | 22.2 | 32.1 | +9.9 |
| Qwen2.5VL-72b | 31.7 | 32.6 | +0.9 |
| Qwen2.5VL-32b | 21.9 | 32.8 | +10.9 |
| ERNIE-4.5-turbo-vl | 32.2 | 33.0 | +0.8 |
| GLM-4.1v-thinking-flashx | 24.6 | 36.0 | +11.4 |
| Doubao-1.5-vision-pro | 42.0 | 49.6 | +7.6 |
| o4-mini | 41.4 | 50.8 | +9.4 |
| Gemini-2.5-flash-thinking | 54.5 | 51.1 | -3.4 |
| o3 | 40.7 | 53.1 | +12.4 |
| Gemini-2.5-pro-thinking | 56.3 | 56.3 | +0.0 |
| Doubao-seed-1.6-thinking | 47.8 | 57.1 | +9.3 |
| Doubao-1.5-thinking-vision-pro | 62.9 | 59.9 | -3.0 |
| Doubao-seed-1.6 | 56.2 | **63.6** | +7.4 |

reaches 34.6, while the best open-source model, InternVL3-8B, also achieves 34.6. These results indicate that multiple-choice questions are more vision-centric, favoring strong visual encoders capable of distinguishing among distractors rather than relying heavily on long-chain reasoning.

**Fill-in-the-blank.** This type yields the highest overall scores, with Doubao-1.5-thinking-vision-pro achieving 67.7 and Doubao-seed-1.6 close behind at 63.8. The best open-source model, ERNIE-4.5-Turbo-VL-Preview, reaches 34.5, and the top reasoning model, WeThink, achieves 30.9. Compared with multiple-choice, fill-in-the-blank questions reward coherent step-by-step reasoning and numerical computation, allowing models with strong symbolic reasoning capabilities to narrow the gap with top vision models. Accuracy in this category could be further improved through better normalization of numeric outputs, unit handling, and formatting.

**Constructed-response.** Performance is moderate, with the top closed-source vision model Doubao-1.5-thinking-vision-pro achieving 51.8, and the best open-source model ERNIE-4.5-Turbo-VL-Preview reaching 29.9. The strongest reasoning-oriented model, MiMo-VL-7B-RL, scores 21.7. Constructed-response questions require multi-step reasoning and coherent explanations, favoring models that can maintain complete reasoning chains and produce structured final answers. Further improvements could be achieved by explicitly presenting intermediate variables and incorporating step verification to reduce omissions.

**Cross-type comparison.** Considering Acc Avg across the three types, the achievable performance ceiling follows the order: Fill-in-the-blank (approximately 68%) ¿ Constructed-response (approximately 53%) ¿ Multiple-choice (approximately 42%). Multiple-choice questions are more dependent on visual recognition, while fill-in-the-blank and constructed-response formats rely more heavily on symbolic reasoning and structured output. Open-source and reasoning-oriented models consistently trail behind the top closed-source models, highlighting gaps in both robust visual encoding and end-to-end reasoning consistency.

### E.2 INTRA-FAMILY PERFORMANCE PATTERNS

The Doubao family demonstrates strong geometric and structured reasoning capabilities. Doubao-1.5-thinking-vision-pro achieves the highest strict accuracy in PG (43.3%), SG (43.2%), and SC (48.5%), indicating superior performance in tasks requiring spatial understanding and formal visual parsing. Within the family, Doubao-seed-1.6 outperforms its thinking variant on more abstract reasoning tasks. In LR, the non-thinking version leads with 32.6%, while the thinking model drops to 17.4%, suggesting that longer reasoning chains may hinder performance under noisy visuals. The Gemini family also shows consistently strong and balanced performance. Gemini-2.5-pro-thinking ranks among the top across tasks, with 48.5% in SC and over 40% in PG and SG. Even in the most challenging LR category, it reaches 39.1%, indicating stable multimodal reasoning. InternVL models show a reversed scaling pattern. The InternVL-3-78B model achieves the best LR score among open models (15.2%), but underperforms the InternVL-3-38B model in SC, possibly due to overfitting or degraded visual generalization at scale. The Qwen2.5VL family excels at structured visual tasks. The 32B model leads in FG (18.6%) and SC (30.3%), showing strength in visual-text alignment. However, scaling to 72B yields only marginal gains, especially in complex reasoning. Overall, different model families show strengths in specific task types—some favor spatial or symbolic inference, others visual parsing. No model excels across all categories, underscoring the current limitations in developing truly general-purpose MLLMs capable of handling diverse visual reasoning tasks.

### E.3 STRICT EVALUATION REVEALS INSTABILITY IN MULTI-STEP REASONING

While many models perform decently under **Acc**, real-world applications often demand fully correct multi-step solutions. Our evaluation reveals clear gaps between $\text{Acc}_{str}$ and Acc, exposing weaknesses in reasoning stability and compositional understanding. For example, Gemini-2.5-pro-thinking scores 48.1% Acc but drops to 42.9% under strict evaluation, reflecting small reasoning failures or incomplete logic. More noticeably, InternVL-3-14B achieves 19.0% Acc but only 10.9% $\text{Acc}_{str}$, a gap of over 8 points, highlighting its difficulty with full-task consistency. Strict metrics thus better reflect whether models can fully solve multi-step problems. They uncover bottlenecks in long-form reasoning and align more closely with educational standards. Reporting both scores is essential for a clearer picture of true problem-solving ability.

### E.4 ANALYSIS OF OCR ACCURACY AND ANSWER ACCURACY

**Overall Performance and Ranking.** Based on Table 8, in the Clean setting the overall accuracy shows a clear gap between the top performers and the rest. Doubao-seed-1.6 ranks first (63.6), followed by Doubao-1.5-thinking-vision-pro (59.9), Gemini-2.5-pro-thinking (56.3), o3 (53.1), Gemini-2.5-flash-thinking (51.1), and o4-mini (50.8). In the Real setting, the best-performing model changes to Doubao-1.5-thinking-vision-pro (62.9), followed by Gemini-2.5-pro-thinking (56.3), Doubao-seed-1.6 (56.2), and Gemini-2.5-flash-thinking (54.5). This indicates that the Doubao family consistently dominates in both conditions, Gemini-2.5-pro-thinking maintains balanced perfor-

Table 9: Comparison of model performances across five categories on multiple-choice questions. PG: Plane Geometry, SG: Solid Geometry, LR: Logical Reasoning, FG: Function Graphs, SC: Statistical Charts. Acc is loose accuracy. The **first** and second highest accuracy of LLMs are bolded and underlined, respectively.

| Model | Acc | | | | | |
|---|---|---|---|---|---|---|
| | PG | SG | LR | FG | SC | **Avg** |
| *LLMs* (Question Text + Figure Description, CoT with 0-shot) | | | | | | |
| Qwen3-235B-A22B-thinking | 12.5 | 60.0 | 66.7 | 14.3 | 66.7 | 34.6 |
| DeepSeek-V3 | 12.5 | 40.0 | 66.7 | 14.3 | 66.7 | 30.8 |
| Qwen3-235B-A22B-instruct | 12.5 | 33.4 | 33.3 | 28.6 | 33.3 | 25.7 |
| DeepSeek-R1 | 25.0 | 60.0 | 66.7 | 14.3 | 66.7 | 38.5 |
| *Closed Models* (Image-only, CoT with 0-shot) | | | | | | |
| Grok-4 | 0.0 | 0.0 | 0.0 | 0.0 | 0.0 | 0.0 |
| Claude-sonnet-4 | 0.0 | 20.0 | 0.0 | 28.6 | 33.3 | 15.4 |
| Claude-sonnet-4-thinking | 0.0 | 0.0 | 0.0 | 14.3 | 66.7 | 11.5 |
| GPT-4.1 | 0.0 | 20.0 | 33.3 | 28.6 | 33.3 | 19.2 |
| GPT-4o | 12.5 | 0.0 | 0.0 | 28.6 | 33.3 | 15.4 |
| Qwen-VL-Max | 0.0 | 0.0 | 0.0 | 28.6 | 33.3 | 11.5 |
| o4-mini | 0.0 | 0.0 | 0.0 | 0.0 | 33.3 | 3.8 |
| o3 | 12.5 | 20.0 | 0.0 | 14.3 | 33.3 | 15.4 |
| Doubao-1.5-vision-pro-32k | 12.5 | 0.0 | 0.0 | 14.3 | 33.3 | 11.5 |
| Doubao-seed-1.6-thinking | 25.0 | 20.0 | 33.3 | 42.9 | 33.3 | 30.8 |
| Gemini-2.5-flash-thinking | 25.0 | 0.0 | 33.3 | 42.9 | 0.0 | 23.1 |
| Gemini-2.5-pro-thinking | 25.0 | 20.0 | 100.0 | 28.6 | 33.3 | 34.6 |
| Doubao-seed-1.6 | 37.5 | 40.0 | 66.7 | 28.6 | 66.7 | **42.3** |
| Doubao-1.5-thinking-vision-pro | 25.0 | 40.0 | 0.0 | 14.3 | 22.3 | 21.8 |
| *Open-source MLLMs* (Image-only, CoT with 0-shot) | | | | | | |
| Gemma-3-4b-it | 0.0 | 0.0 | 0.0 | 0.0 | 0.0 | 0.0 |
| Gemma-3n-E4B | 0.0 | 0.0 | 33.3 | 42.9 | 0.0 | 15.4 |
| Gemma-3-27b-it | 12.5 | 0.0 | 33.3 | 14.3 | 0.0 | 11.5 |
| Kimi-VL-A3B-Instruct | 0.0 | 0.0 | 0.0 | 28.6 | 0.0 | 7.7 |
| Qwen2.5-VL-7B-Instruct | 0.0 | 20.0 | 0.0 | 14.3 | 0.0 | 7.7 |
| InternVL3-8B | 37.5 | 20.0 | 0.0 | 42.9 | 66.7 | **34.6** |
| InternVL3-14B | 25.0 | 0.0 | 33.3 | 14.3 | 100.0 | 26.9 |
| Llama-4-Maverick | 0.0 | 0.0 | 0.0 | 28.6 | 33.3 | 11.5 |
| InternVL3-78B | 12.5 | 0.0 | 0.0 | 14.3 | 0.0 | 7.7 |
| Qwen2.5-VL-32B-Instruct | 12.5 | 0.0 | 33.3 | 28.6 | 33.3 | 19.2 |
| InternVL3-38B | 12.5 | 20.0 | 0.0 | 42.9 | 66.7 | 26.9 |
| GLM-4.1v-thinking-flashx | 0.0 | 0.0 | 33.3 | 28.6 | 33.3 | 15.4 |
| Qwen2.5-VL-72B | 12.5 | 0.0 | 0.0 | 42.9 | 33.3 | 19.2 |
| ERNIE-4.5-Turbo-VL-Preview | 25.0 | 0.0 | 0.0 | 28.6 | 33.3 | 19.2 |
| *Reasoner* (Image-only, CoT with 0-shot) | | | | | | |
| Keye-VL-8B-Preview | 0.0 | 0.0 | 0.0 | 14.3 | 33.3 | 7.7 |
| OVR | 0.0 | 0.0 | 0.0 | 0.0 | 66.7 | 7.7 |
| Revisual-R1 | 0.0 | 0.0 | 0.0 | 14.3 | 66.7 | 11.5 |
| OpenVLThinker | 0.0 | 0.0 | 0.0 | 28.6 | 0.0 | 7.7 |
| ThinkLite-VL | 25.0 | 0.0 | 0.0 | 14.3 | 33.3 | 15.4 |
| VLAA-Thinker-Qwen2.5VL-7B | 12.5 | 20.0 | 0.0 | 14.3 | 0.0 | 11.5 |
| WeThink | 12.5 | 0.0 | 0.0 | 14.3 | 33.3 | 11.5 |
| MMR1-Math-v0-7B | 12.5 | 20.0 | 0.0 | 14.3 | 33.3 | 15.4 |
| MM-Eureka | 12.5 | 20.0 | 0.0 | 14.3 | 55.7 | 18.0 |
| MiMo-VL-7B-RL | 0.0 | 0.0 | 0.0 | 0.0 | 33.3 | 3.8 |
| VL-Rethinker-7B | 25.0 | 40.0 | 0.0 | 28.6 | 66.7 | **30.8** |
| Skywork-R1V3-38B | 25.0 | 20.0 | 33.3 | 14.3 | 77.7 | 28.2 |

Table 10: Comparison of model performances across five categories on fill-in-the-blank questions. PG: Plane Geometry, SG: Solid Geometry, LR: Logical Reasoning, FG: Function Graphs, SC: Statistical Charts. $Acc_{str}$ is strict accuracy, Acc is loose accuracy. The **first** and second highest accuracy of LLMs are bolded and underlined, respectively.

| Model | $Acc_{str}$ | | | | | | Acc | | | | | |
|---|---|---|---|---|---|---|---|---|---|---|---|---|
| | PG | SG | LR | FG | SC | Avg | PG | SG | LR | FG | SC | Avg |
| *LLMs* (Question Text + Figure Description, CoT with 0-shot) | | | | | | | | | | | | |
| Qwen3-235B-A22B-thinking | 41.5 | 7.1 | 57.1 | 23.1 | 58.3 | 39.8 | 49.4 | 20.9 | 68.9 | 26.9 | 67.3 | 48.8 |
| DeepSeek-V3 | 37.7 | 35.7 | 38.1 | 30.8 | 50.0 | 38.1 | 47.2 | 44.0 | 51.5 | 53.9 | 60.4 | 49.8 |
| Qwen3-235B-A22B-instruct | 47.2 | 21.4 | 38.1 | 30.8 | 50.0 | 40.7 | 60.0 | 36.1 | 60.6 | 46.8 | 62.4 | 55.9 |
| DeepSeek-R1 | 49.1 | 50.0 | 38.1 | 23.1 | 50.0 | 44.2 | 60.3 | 55.9 | 50.6 | 56.5 | 74.3 | 59.0 |
| *Closed Models* (Image-only, CoT with 0-shot) | | | | | | | | | | | | |
| Grok-4 | 11.3 | 7.1 | 0.0 | 0.0 | 0.0 | 6.2 | 16.8 | 9.5 | 6.3 | 0.0 | 6.2 | 10.9 |
| Claude-sonnet-4 | 11.3 | 7.1 | 14.3 | 0.0 | 8.3 | 9.7 | 19.2 | 14.2 | 19.0 | 20.6 | 27.8 | 19.6 |
| Claude-sonnet-4-thinking | 18.9 | 7.1 | 19.0 | 0.0 | 8.3 | 14.2 | 30.2 | 16.6 | 20.6 | 20.5 | 25.7 | 25.2 |
| GPT-4.1 | 17.0 | 14.3 | 14.3 | 15.4 | 25.0 | 16.8 | 23.7 | 14.3 | 28.5 | 28.2 | 45.2 | 26.2 |
| GPT-4o | 18.9 | 14.3 | 14.3 | 7.7 | 0.0 | 14.2 | 26.5 | 22.0 | 31.3 | 25.6 | 28.5 | 27.0 |
| Qwen-VL-Max | 17.0 | 35.7 | 14.3 | 30.8 | 41.7 | 23.0 | 24.6 | 41.4 | 23.8 | 43.6 | 58.4 | 32.3 |
| o4-mini | 30.2 | 28.6 | 33.3 | 30.8 | 16.7 | 29.2 | 41.2 | 35.7 | 46.0 | 53.2 | 34.1 | 42.1 |
| o3 | 43.4 | 42.9 | 23.8 | 38.5 | 25.0 | 37.2 | 60.3 | 52.4 | 36.6 | 55.2 | 43.8 | 52.6 |
| Doubao-1.5-vision-pro-32k | 28.3 | 35.7 | 23.8 | 7.7 | 16.7 | 24.8 | 40.7 | 40.4 | 41.3 | 38.4 | 53.4 | 41.9 |
| Doubao-seed-1.6-thinking | 47.2 | 50.0 | 23.8 | 30.8 | 25.0 | 38.9 | 60.0 | 59.5 | 40.8 | 53.8 | 58.2 | 55.5 |
| Gemini-2.5-flash-thinking | 50.9 | 57.1 | 28.6 | 38.5 | 41.7 | 45.1 | 62.2 | 61.9 | 46.0 | 52.5 | 70.8 | 59.0 |
| Gemini-2.5-pro-thinking | 45.3 | 42.9 | 47.6 | 30.8 | 50.0 | 44.2 | 57.5 | 63.5 | 58.7 | 42.9 | 74.3 | 58.6 |
| Doubao-seed-1.6 | 50.9 | 57.1 | 52.4 | 30.8 | 33.3 | 47.8 | 60.1 | 66.7 | 73.2 | 51.2 | 74.0 | 63.8 |
| Doubao-1.5-thinking-vision-pro | 58.5 | 42.9 | 38.1 | 30.8 | 58.3 | **49.6** | 71.2 | 65.9 | 56.6 | 59.6 | 82.0 | **67.7** |
| *Open-source MLLMs* (Image-only, CoT with 0-shot) | | | | | | | | | | | | |
| Gemma-3-4b-it | 3.8 | 0.0 | 0.0 | 0.0 | 0.0 | 1.8 | 6.5 | 2.4 | 0.0 | 0.0 | 2.8 | 3.6 |
| Gemma-3n-E4B | 7.5 | 7.1 | 0.0 | 0.0 | 0.0 | 4.4 | 13.5 | 17.9 | 9.8 | 8.3 | 16.7 | 13.1 |
| Gemma-3-27b-it | 3.8 | 0.0 | 0.0 | 0.0 | 0.0 | 1.8 | 9.6 | 7.1 | 8.7 | 12.8 | 13.8 | 9.9 |
| Kimi-VL-A3B-Instruct | 7.5 | 14.3 | 0.0 | 7.7 | 0.0 | 6.2 | 16.9 | 16.6 | 17.4 | 18.0 | 8.2 | 16.2 |
| Qwen2.5-VL-7B-Instruct | 3.8 | 28.6 | 19.0 | 7.7 | 16.7 | 11.5 | 17.5 | 36.3 | 26.1 | 29.5 | 31.2 | 24.3 |
| InternVL3-8B | 9.4 | 14.3 | 0.0 | 0.0 | 0.0 | 6.2 | 18.0 | 22.6 | 9.0 | 15.4 | 25.3 | 17.4 |
| InternVL3-14B | 7.5 | 21.4 | 9.5 | 7.7 | 8.3 | 9.7 | 16.8 | 28.6 | 23.1 | 24.3 | 29.9 | 21.7 |
| Llama-4-Maverick | 17.0 | 14.3 | 19.0 | 7.7 | 0.0 | 14.2 | 26.2 | 22.6 | 25.3 | 15.4 | 20.8 | 23.8 |
| InternVL3-78B | 9.4 | 35.7 | 14.3 | 23.1 | 16.7 | 15.9 | 18.8 | 38.1 | 24.6 | 46.1 | 41.4 | 27.8 |
| Qwen2.5-VL-32B-Instruct | 9.4 | 35.7 | 14.3 | 30.8 | 33.3 | **18.6** | 16.8 | 38.1 | 24.5 | 53.9 | 50.0 | 28.7 |
| InternVL3-38B | 17.0 | 28.6 | 4.8 | 7.7 | 16.7 | 15.0 | 25.4 | 33.4 | 17.5 | 27.6 | 37.5 | 26.4 |
| GLM-4.1v-thinking-flashx | 13.2 | 0.0 | 9.5 | 15.4 | 8.3 | 10.6 | 27.5 | 20.6 | 22.2 | 31.4 | 34.0 | 26.8 |
| Qwen2.5-VL-72B | 13.2 | 21.4 | 14.3 | 7.7 | 8.3 | 13.3 | 25.2 | 42.6 | 30.0 | 38.4 | 54.2 | 32.8 |
| ERNIE-4.5-Turbo-VL-Preview | 20.8 | 21.4 | 9.5 | 15.4 | 25.0 | **18.6** | 30.7 | 38.6 | 23.8 | 37.8 | 61.6 | **34.5** |
| *Reasoner* (Image-only, CoT with 0-shot) | | | | | | | | | | | | |
| Keye-VL-8B-Preview | 3.8 | 7.1 | 0.0 | 0.0 | 0.0 | 2.7 | 4.9 | 7.1 | 1.6 | 0.0 | 17.3 | 5.3 |
| OVR | 1.9 | 7.1 | 4.8 | 7.7 | 8.3 | 4.4 | 5.0 | 7.1 | 9.5 | 20.5 | 15.0 | 9.0 |
| Revisual-R1 | 5.7 | 14.3 | 4.8 | 0.0 | 0.0 | 5.3 | 14.8 | 14.3 | 9.5 | 5.2 | 17.4 | 12.9 |
| OpenVLThinker | 13.2 | 21.4 | 4.8 | 15.4 | 16.7 | 13.3 | 19.0 | 38.6 | 18.9 | 33.3 | 29.8 | 24.2 |
| ThinkLite-VL | 9.4 | 28.6 | 14.3 | 7.7 | 8.3 | 12.4 | 17.4 | 38.7 | 25.3 | 26.2 | 38.2 | 24.8 |
| VLAA-Thinker-Qwen2.5VL-7B | 5.7 | 14.3 | 4.8 | 15.4 | 0.0 | 7.1 | 16.7 | 26.8 | 16.0 | 42.3 | 39.4 | 23.2 |
| WeThink | 7.5 | 21.4 | 19.0 | 23.1 | 8.3 | 13.3 | 20.8 | 37.1 | 36.8 | 38.4 | 49.8 | 30.9 |
| MMR1-Math-v0-7B | 5.7 | 14.3 | 4.8 | 15.4 | 8.3 | 8.0 | 20.2 | 16.6 | 20.1 | 34.5 | 45.1 | 24.1 |
| MM-Eureka | 7.5 | 28.6 | 9.5 | 7.7 | 0.0 | 9.7 | 24.2 | 33.4 | 20.3 | 33.3 | 38.9 | 27.2 |
| MiMo-VL-7B-RL | 18.9 | 14.3 | 9.5 | 7.7 | 16.7 | 15.0 | 28.0 | 21.4 | 15.9 | 23.7 | 44.4 | 26.2 |
| VL-Rethinker-7B | 11.3 | 21.4 | 14.3 | 15.4 | 8.3 | 13.3 | 26.0 | 33.9 | 29.7 | 35.9 | 41.0 | 30.4 |
| Skywork-R1V3-38B | 20.8 | 7.1 | 9.5 | 30.8 | 8.3 | **16.8** | 35.8 | 30.4 | 19.8 | 52.5 | 27.7 | **33.2** |

Table 11: Comparison of model performances across five categories on constructed-response questions. PG: Plane Geometry, SG: Solid Geometry, LR: Logical Reasoning, FG: Function Graphs, SC: Statistical Charts. $Acc_{str}$ is strict accuracy, Acc is loose accuracy. The **first** and second highest accuracy of LLMs are bolded and underlined, respectively.

| Model | $Acc_{str}$ | | | | | | Acc | | | | | |
|---|---|---|---|---|---|---|---|---|---|---|---|---|
| | PG | SG | LR | FG | SC | Avg | PG | SG | LR | FG | SC | Avg |
| *LLMs* (Question Text + Figure Description, CoT with 0-shot) | | | | | | | | | | | | |
| Qwen3-235B-A22B-thinking | 26.3 | 32.6 | 22.7 | 21.7 | 38.9 | 28.2 | 32.1 | 37.5 | 27.3 | 31.9 | 56.5 | 34.5 |
| DeepSeek-V3 | 25.3 | 30.4 | 27.3 | 30.4 | 61.1 | 29.0 | 42.4 | 35.1 | 39.8 | 42.4 | 76.8 | 42.1 |
| Qwen3-235B-A22B-instruct | 31.2 | 35.9 | 36.4 | 47.8 | 44.4 | 34.6 | 43.4 | 42.0 | 44.0 | 62.7 | 63.8 | 45.4 |
| DeepSeek-R1 | 41.9 | 33.7 | 40.9 | 39.1 | 61.1 | 40.5 | 56.6 | 42.0 | 50.7 | 52.9 | 80.6 | 53.3 |
| *Closed Models* (Image-only, CoT with 0-shot) | | | | | | | | | | | | |
| Grok-4 | 4.3 | 2.2 | 0.0 | 0.0 | 0.0 | 2.9 | 5.5 | 3.3 | 0.0 | 0.0 | 1.8 | 4.0 |
| Claude-sonnet-4 | 6.5 | 6.5 | 4.5 | 0.0 | 16.7 | 6.5 | 13.6 | 8.2 | 12.1 | 17.8 | 26.4 | 13.0 |
| Claude-sonnet-4-thinking | 9.1 | 7.6 | 4.5 | 13.0 | 11.1 | 8.8 | 16.8 | 8.3 | 12.1 | 14.5 | 14.8 | 14.0 |
| GPT-4.1 | 11.3 | 14.1 | 9.1 | 0.0 | 33.3 | 12.3 | 21.1 | 19.5 | 18.9 | 19.6 | 43.5 | 21.6 |
| GPT-4o | 11.8 | 15.2 | 13.6 | 8.7 | 22.2 | 13.2 | 22.7 | 20.8 | 21.2 | 22.5 | 25.9 | 22.2 |
| Qwen-VL-Max | 9.1 | 10.9 | 9.1 | 4.3 | 22.2 | 10.0 | 21.4 | 17.8 | 19.7 | 25.4 | 25.9 | 20.8 |
| o4-mini | 26.3 | 23.9 | 13.6 | 17.4 | 33.3 | 24.6 | 37.8 | 30.0 | 20.1 | 36.3 | 48.2 | 35.0 |
| o3 | 23.1 | 28.3 | 9.1 | 0.0 | 44.4 | 23.2 | 31.9 | 34.5 | 19.7 | 11.7 | 46.3 | 31.2 |
| Doubao-1.5-vision-pro-32k | 28.0 | 28.3 | 18.2 | 30.4 | 33.3 | 27.9 | 42.6 | 38.2 | 24.3 | 47.9 | 37.0 | 40.3 |
| Doubao-seed-1.6-thinking | 34.4 | 23.9 | 9.1 | 43.5 | 33.3 | 30.5 | 46.1 | 30.5 | 21.2 | 49.3 | 57.8 | 41.1 |
| Gemini-2.5-flash-thinking | 41.4 | 35.9 | 13.6 | 43.5 | 61.1 | 39.3 | 53.2 | 42.6 | 27.3 | 53.7 | 70.8 | 49.6 |
| Gemini-2.5-pro-thinking | 39.2 | 42.4 | 22.7 | 47.8 | 50.0 | **40.2** | 50.7 | 47.3 | 34.9 | 60.2 | 59.7 | 49.8 |
| Doubao-seed-1.6 | 38.2 | 34.8 | 9.1 | 43.5 | 55.6 | 36.7 | 51.7 | 41.9 | 24.6 | 55.4 | 59.2 | 48.0 |
| Doubao-1.5-thinking-vision-pro | 39.8 | 43.5 | 18.2 | 39.1 | 50.0 | 39.9 | 53.2 | 50.6 | 31.8 | 55.1 | 63.9 | **51.8** |
| *Open-source MLLMs* (Image-only, CoT with 0-shot) | | | | | | | | | | | | |
| Gemma-3-4b-it | 0.5 | 2.2 | 4.5 | 0.0 | 0.0 | 1.2 | 3.7 | 2.5 | 6.0 | 0.0 | 0.0 | 3.1 |
| Gemma-3n-E4B | 1.1 | 2.2 | 4.5 | 0.0 | 11.1 | 2.1 | 6.8 | 5.3 | 9.1 | 2.9 | 17.1 | 6.8 |
| Gemma-3-27b-it | 4.3 | 5.4 | 0.0 | 0.0 | 11.1 | 4.4 | 10.0 | 6.2 | 3.0 | 6.1 | 14.8 | 8.5 |
| Kimi-VL-A3B-Instruct | 2.7 | 10.9 | 0.0 | 4.3 | 0.0 | 4.7 | 10.0 | 14.9 | 3.0 | 14.5 | 11.6 | 11.3 |
| Qwen2.5-VL-7B-Instruct | 4.3 | 5.4 | 9.1 | 0.0 | 0.0 | 4.4 | 14.9 | 12.5 | 13.1 | 13.1 | 19.0 | 14.5 |
| InternVL3-8B | 7.0 | 9.8 | 9.1 | 4.3 | 11.1 | 7.9 | 14.5 | 15.4 | 15.1 | 7.2 | 27.3 | 15.0 |
| InternVL3-14B | 7.0 | 14.1 | 4.5 | 0.0 | 16.7 | 8.8 | 14.8 | 18.2 | 15.1 | 8.7 | 28.2 | 16.1 |
| Llama-4-Maverick | 10.2 | 10.9 | 9.1 | 4.3 | 5.6 | 9.7 | 18.9 | 13.3 | 21.2 | 17.4 | 21.8 | 17.6 |
| InternVL3-78B | 7.0 | 13.0 | 18.2 | 4.3 | 16.7 | 9.7 | 17.1 | 17.2 | 27.3 | 17.4 | 35.6 | 18.8 |
| Qwen2.5-VL-32B-Instruct | 8.6 | 10.9 | 9.1 | 8.7 | 27.8 | 10.3 | 19.1 | 16.4 | 13.6 | 20.3 | 37.0 | 19.0 |
| InternVL3-38B | 8.1 | 14.1 | 13.6 | 4.3 | 22.2 | 10.6 | 18.1 | 17.6 | 16.6 | 20.3 | 41.2 | 19.2 |
| GLM-4.1v-thinking-flashx | 15.1 | 15.2 | 4.5 | 0.0 | 22.2 | 13.8 | 28.2 | 21.7 | 7.6 | 16.0 | 31.4 | 24.4 |
| Qwen2.5-VL-72B | 12.4 | 17.4 | 9.1 | 13.0 | 22.2 | 14.1 | 27.5 | 22.6 | 13.6 | 30.4 | 35.7 | 25.9 |
| ERNIE-4.5-Turbo-VL-Preview | 17.2 | 13.0 | 18.2 | 13.0 | 27.8 | **16.4** | 33.3 | 20.1 | 28.8 | 31.2 | 45.3 | **29.9** |
| *Reasoner* (Image-only, CoT with 0-shot) | | | | | | | | | | | | |
| Keye-VL-8B-Preview | 3.2 | 4.3 | 0.0 | 4.3 | 5.6 | 3.5 | 4.8 | 4.7 | 0.0 | 4.3 | 7.4 | 4.6 |
| OVR | 3.2 | 5.4 | 4.5 | 8.7 | 11.1 | 4.7 | 7.6 | 7.6 | 10.6 | 16.3 | 14.8 | 8.7 |
| Revisual-R1 | 6.5 | 5.4 | 4.5 | 4.3 | 11.1 | 6.2 | 11.6 | 6.9 | 9.1 | 10.1 | 25.0 | 10.8 |
| OpenVLThinker | 3.2 | 7.6 | 9.1 | 0.0 | 11.1 | 5.0 | 14.2 | 11.5 | 9.1 | 11.6 | 25.4 | 13.6 |
| ThinkLite-VL | 4.3 | 7.6 | 4.5 | 0.0 | 11.1 | 5.3 | 16.1 | 12.5 | 9.1 | 18.5 | 28.7 | 15.5 |
| VLAA-Thinker-Qwen2.5VL-7B | 5.4 | 9.8 | 13.6 | 0.0 | 16.7 | 7.3 | 16.0 | 16.1 | 22.7 | 14.5 | 36.1 | 17.4 |
| WeThink | 6.5 | 8.7 | 9.1 | 4.3 | 5.6 | 7.0 | 16.8 | 16.2 | 21.6 | 18.9 | 22.2 | 17.4 |
| MMR1-Math-v0-7B | 9.7 | 10.9 | 4.5 | 4.3 | 11.1 | 9.4 | 20.1 | 18.0 | 10.6 | 21.8 | 28.7 | 19.5 |
| MM-Eureka | 5.4 | 14.1 | 9.1 | 0.0 | 22.2 | 8.5 | 17.3 | 19.8 | 20.5 | 12.3 | 36.1 | 18.8 |
| MiMo-VL-7B-RL | 15.1 | 13.0 | 0.0 | 13.0 | 22.2 | 13.8 | 23.4 | 19.8 | 6.0 | 20.3 | 33.8 | 21.7 |
| VL-Rethinker-7B | 9.7 | 13.0 | 13.6 | 8.7 | 16.7 | 11.1 | 20.2 | 18.7 | 19.7 | 26.0 | 26.3 | 20.5 |
| Skywork-R1V3-38B | 17.2 | 7.6 | 4.5 | 17.4 | 22.2 | **14.1** | 28.9 | 13.7 | 15.1 | 31.9 | 31.4 | **24.3** |

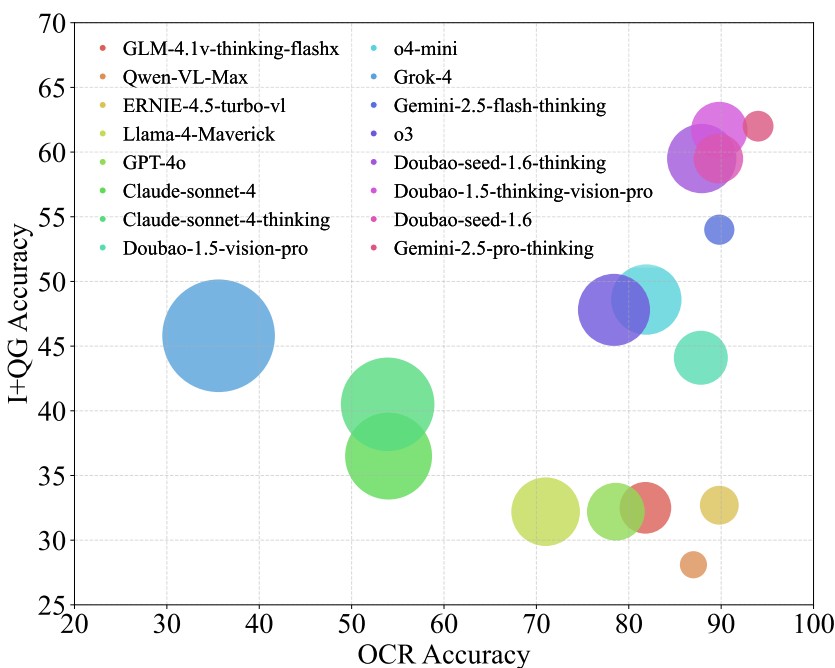

Figure 11: Scatter plot of the relationship between OCR accuracy and accuracy in the I+QG setting, where the size of each circle represents the difference in accuracy between the I+QG setting and the I+QM setting.

mance across domains, while models like o3 and o4-mini have stronger upper bounds in the Clean setting but drop in ranking for Real, showing higher sensitivity to input cleanliness.

**Robustness and $\Delta$ Analysis.** From the perspective of $\Delta = \text{Acc}_{\text{Clean}} - \text{Acc}_{\text{Real}}$, a smaller absolute value indicates greater robustness across domains. The most stable model is Gemini-2.5-pro-thinking ($\Delta = 0.0$), followed by ERNIE-4.5-turbo-vl (+0.8), InternVL3-14b (+0.6), and Qwen2.5VL-72b (+0.9), suggesting minimal dependence on input cleaning. Most mainstream models gain between 5 and 10 percentage points in Clean compared to Real, such as GPT-4o (+6.9), GPT-4.1 (+6.8), o4-mini (+9.4), Doubao-1.5-vision-pro (+7.6), and Qwen-VL-Max (+9.9), indicating that standardization and denoising benefit a wide range of systems. Notably, two atypical patterns emerge: first, models with negative $\Delta$, including Gemini-2.5-flash-thinking (-3.4) and Doubao-1.5-thinking-vision-pro (-3.0), perform better in Real than in Clean, possibly due to stronger adaptation to realistic noise and layout variations; second, models with very large $\Delta$, such as Llama-4-Maverick (+13.3), o3 (+12.4), Claude-sonnet-4-thinking (+11.7), GLM-4.1v-thinking-flashx (+11.4), Qwen2.5VL-32b (+10.9), and Claude-sonnet-4 (+10.9), show substantial benefits from cleaner inputs, implying higher vulnerability to noise and complex formatting.

**Family and Model-Type Comparison.** Within the Doubao series, Doubao-1.5-thinking-vision-pro leads in Real accuracy (62.9) but slightly drops in Clean (negative $\Delta$), making it well-suited for raw, noisy data. Doubao-seed-1.6 achieves the highest Clean score (63.6) while remaining competitive in Real (56.2), representing the strongest all-around performer. The Gemini family presents a contrast: Gemini-2.5-pro-thinking achieves perfect robustness ($\Delta = 0$) and high scores in both domains, while Gemini-2.5-flash-thinking is notably stronger in Real than Clean. OpenAI's o3 and o4-mini benefit greatly from cleaner inputs (large positive $\Delta$), making them excellent candidates for pipelines with strong preprocessing. Other major model families, such as GPT-4o/4.1, Claude, Qwen, and InternVL, generally follow the trend of significantly higher accuracy in Clean, reinforcing the importance of preprocessing for optimal performance.

## F  THE USE OF LARGE LANGUAGE MODELS

In this work, we used LLMs only in a supportive role for aid and polish writing. Specifically, LLM assistance was employed for improving the clarity and fluency of exposition in the Abstract, Introduction, and Related Work sections. In addition, LLMs were used for formatting support, including converting mathematical expressions into standard LaTeX notation and organizing dataset statistics and results into well-formatted tables and figures. All substantive research contributions were performed entirely by the authors without reliance on LLMs.

# 1. Plane Geometry

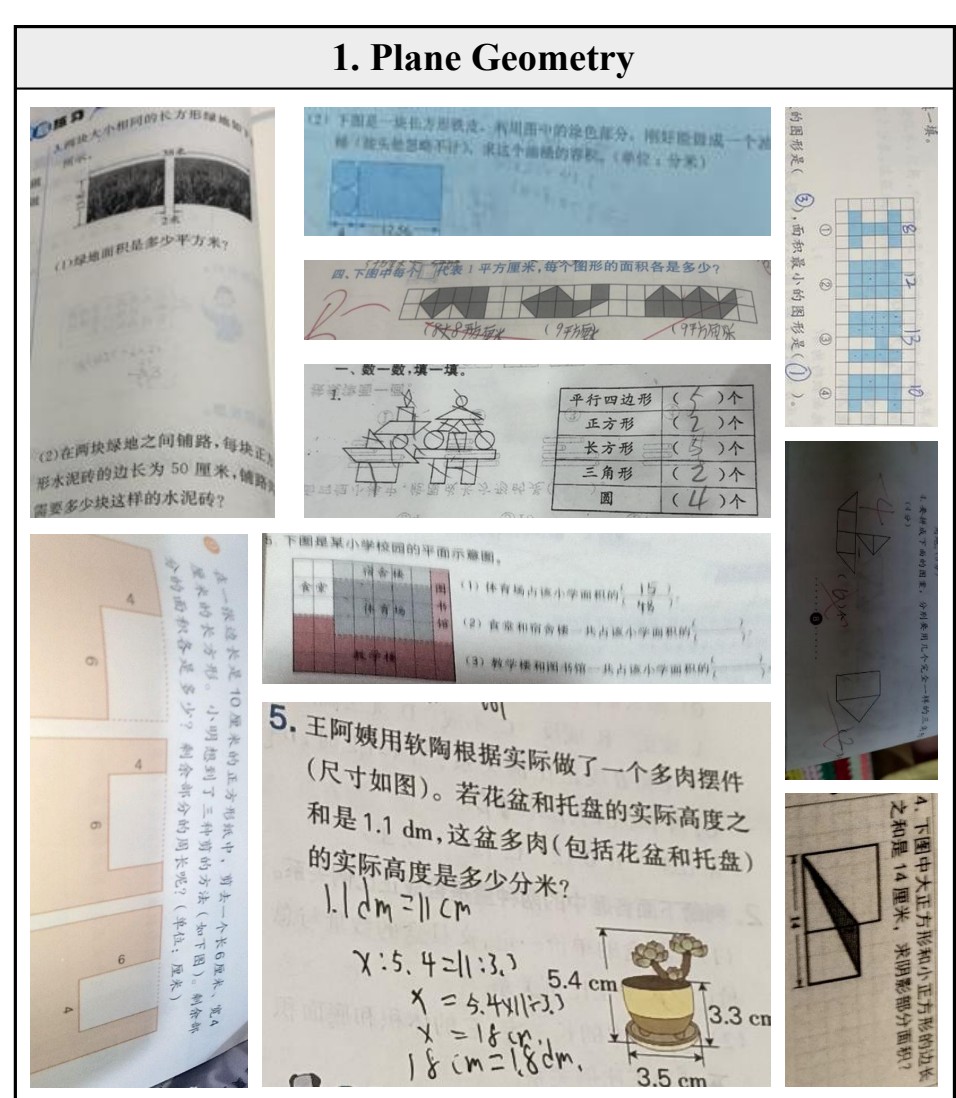

Figure 12: Samples of Plane Geometry.

# 2. Solid Geometry

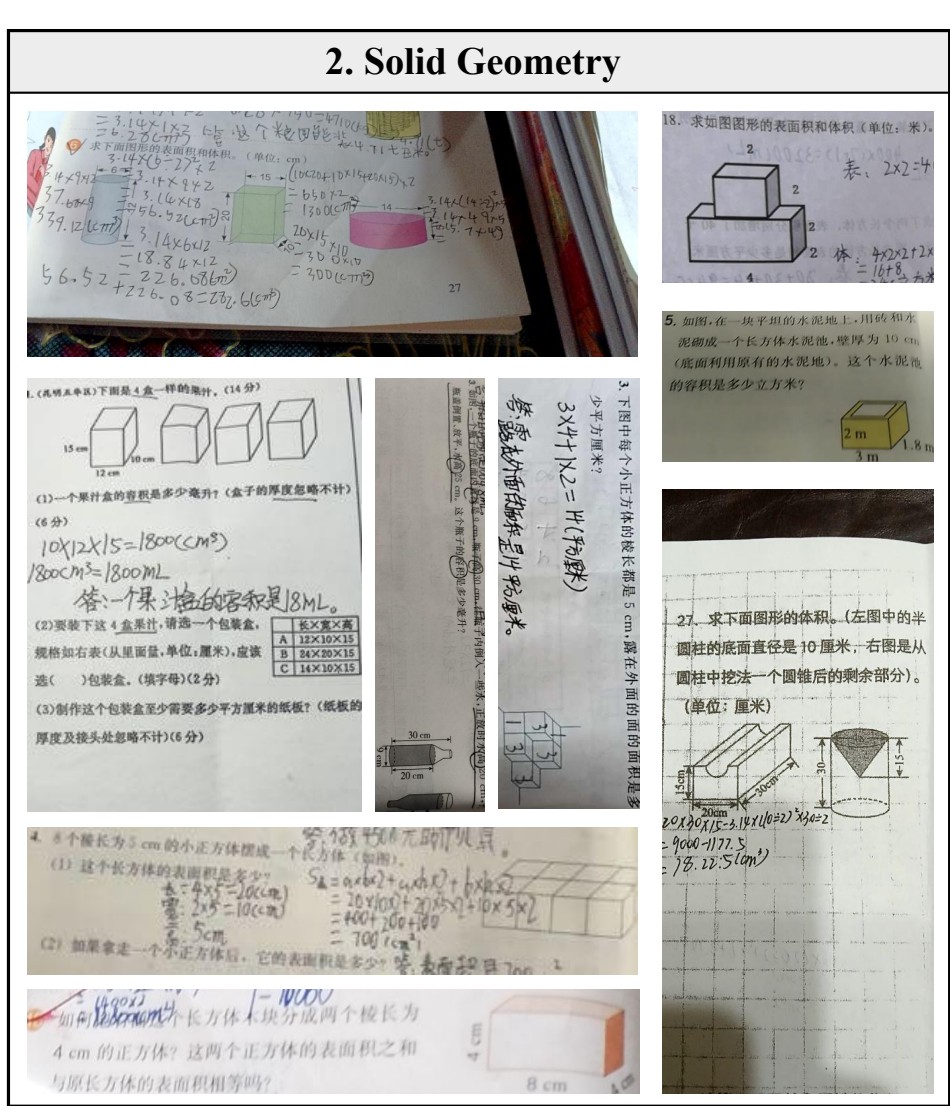

Figure 13: Samples of Solid Geometry.

# 3. Logical Reasoning

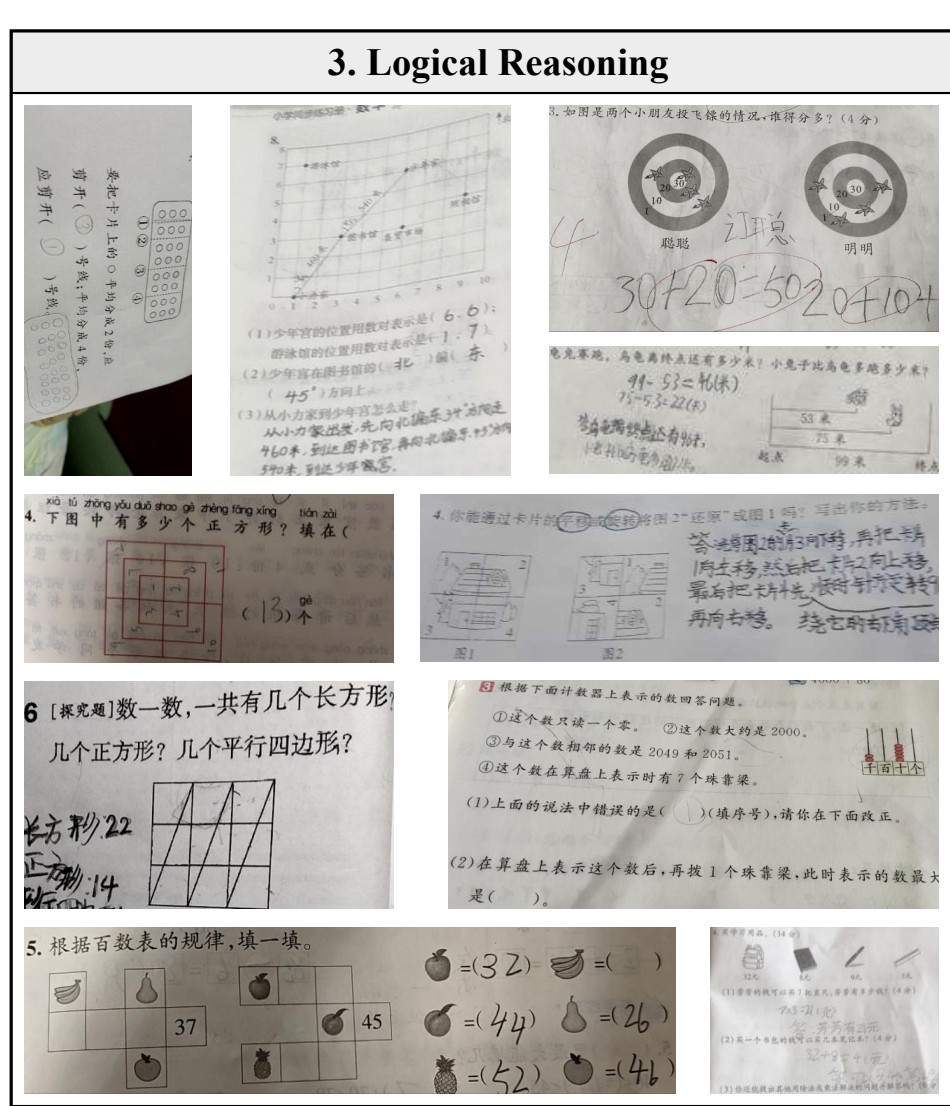

Figure 14: Samples of Logical Reasoning.

## 4. Function Graphs

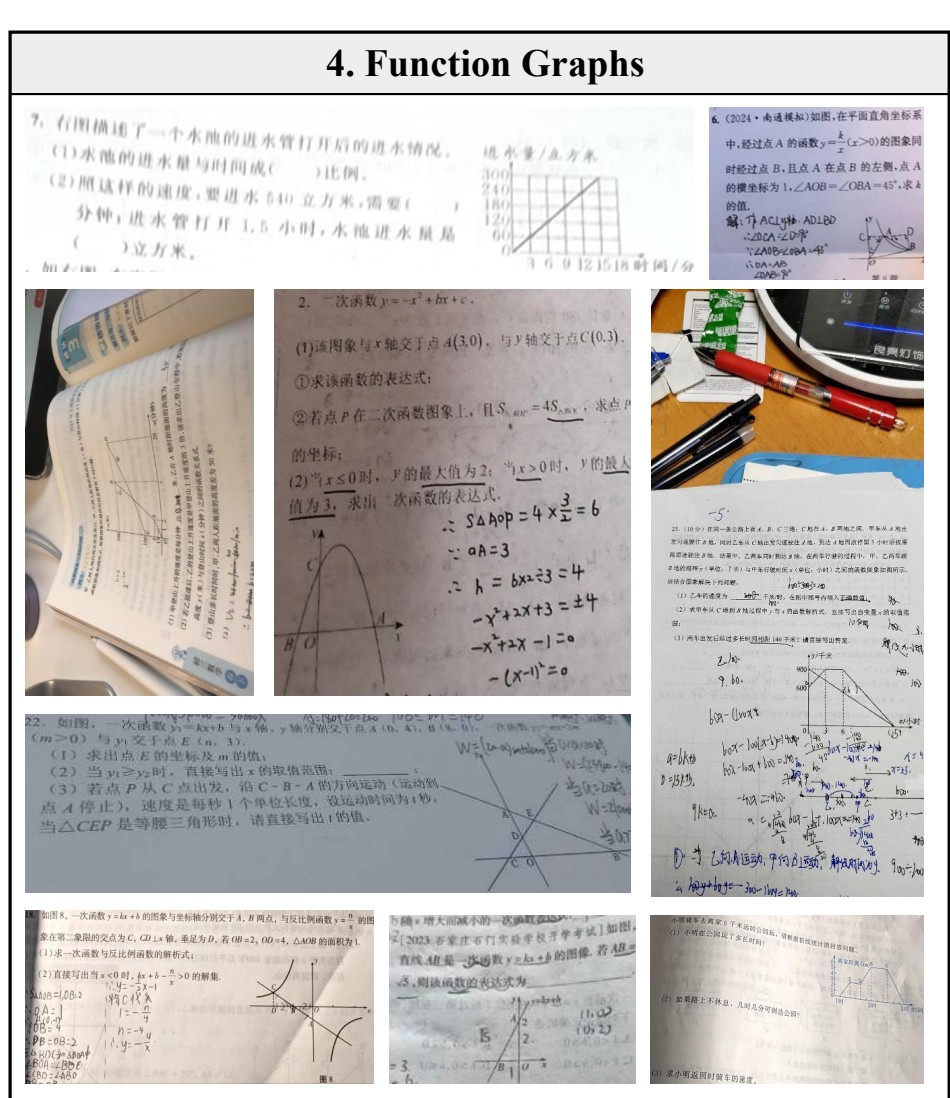

Figure 15: Samples of Function Graphs.

# 5. Statistical Charts

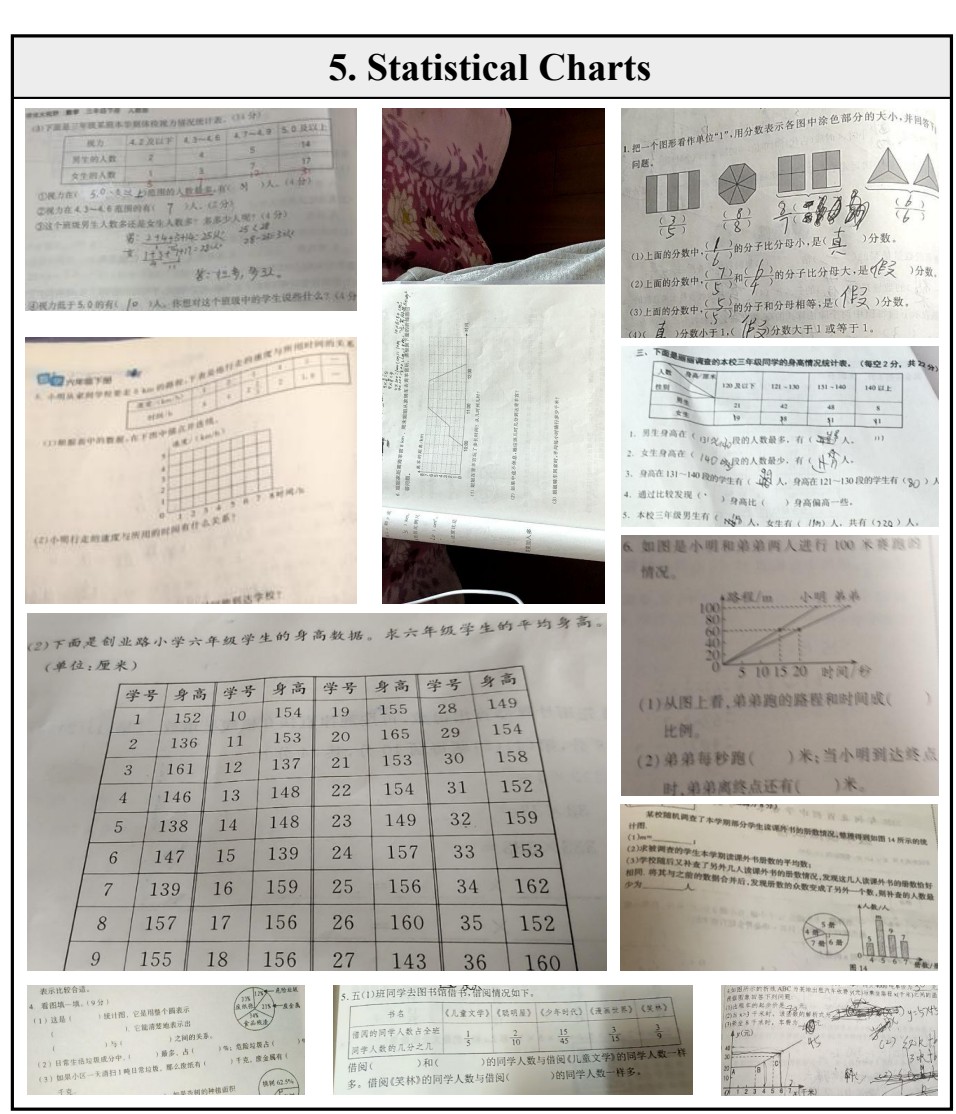

Figure 16: Samples of Statistical Charts.

