# OpenReview forum: "MathReal: We Keep It Real! A Real Scene Benchmark for Evaluating Math Reasoning in Multimodal Large Language Models"
_ICLR.cc/2026/Conference — ICLR 2026 Conference Withdrawn Submission_

### Official Review · Reviewer_7xbv · 2025-10-30

**Soundness:** 3
**Presentation:** 2
**Contribution:** 2
**Rating:** 4
**Confidence:** 4

**Summary:**

This paper focuses on K-12 issues and proposes a benchmark composed of handwritten data captured using mobile devices, making it more realistic. Detailed analysis and extensive experiments demonstrate that MathReal can indeed test large models' ability to recognize real-world images, and that current multimodal large models still have significant difficulty in recognizing noisy images.

**Strengths:**

1. The paper clearly points out that existing mathematical multimodal reasoning benchmarks mainly focus on 'clean image' scenarios and lack tests in real educational settings (problems photographed by K–12 students on their phones), making the motivation reasonable and practically significant.It emphasizes that noise in real images (such as blurriness, perspective changes, and handwritten interference) is indeed a current weakness of MLLMs.

2. Covers 40 MLLMs, including both open-source and closed-source, as well as reasoner-type models, with multiple experimental dimensions. Provides "six input modes" (I, I+QM, I+QG, I+QG+DG, etc.), which help distinguish between OCR and reasoning stages.

**Weaknesses:**

1. All samples come from a K–12 context, and their effectiveness in generalizing to international or higher-level tasks is limited. The benchmark lacks classifications for subjects and fields, as well as the varying impacts of image noise in these areas.
2. Although the distribution of error types is illustrated, there is a lack of quantitative analysis (for example, the contribution of different noise types to OCR errors). What specific noise can make a great difference?
3. Compared with existing visual mathematics benchmarks such as MathVista, MathVerse, and MathGlance, the main innovation lies in the 'real-world image capture' dimension, with moderate academic novelty. As for 'Real Scene', it seems like more of an engineering pipeline rather than a conceptual or algorithmic innovation.

**Questions:**

See Weaknesses.

---

### Official Review · Reviewer_ZutB · 2025-10-30

**Soundness:** 1
**Presentation:** 3
**Contribution:** 2
**Rating:** 2
**Confidence:** 4

**Summary:**

This paper introduces MathReal, a benchmark of 2,000 K–12 mathematical questions captured from real-world educational contexts using handheld devices. Each image contains both the question text and figures, reflecting real visual noise such as blur, rotation, and handwritten markings. The authors argue that previous multimodal math benchmarks rely on clean, synthetic images, and that MathReal provides the first “real-world” benchmark for evaluating MLLMs in authentic educational scenarios. They evaluate 40 models under six experimental settings and conduct detailed error analyses, showing significant performance degradation on real images compared to clean ones.

**Strengths:**

1. The paper tackles a practically meaningful problem—evaluating MLLMs under realistic conditions where image quality and layout are imperfect.
2. The dataset is systematically annotated with visual degradation categories, educational levels, and question types, providing a structured way to analyze failure modes.
3. The experimental coverage is extensive, including 40 models and both open- and closed-source ones, with a consistent evaluation protocol.
4. The error taxonomy (OCR, perception, reasoning, hallucination, etc.) is insightful and helps identify concrete bottlenecks in MLLM pipelines.

**Weaknesses:**

1. The paper overclaims novelty by stating MathReal is “the first real-world benchmark” for visual math reasoning. Similar “real-scene” or “in-the-wild” multimodal math datasets—are introduced in the ACM MM 2025 paper https://dl.acm.org/doi/10.1145/3746027.3758240—already explored authentic photo-based or user-captured math scenarios. These should be cited and compared directly.

2. The dataset scale (2,000 images) is relatively small and may not justify the “benchmark” positioning without stronger generalization or robustness analysis.

3. While the realness aspect is emphasized, there is limited discussion on data diversity (e.g., lighting, device type, geographic/cultural coverage) and license/ethics of data collection.

4. Some analyses (e.g., LLM-as-a-Judge reliability) are standard and not novel; the contribution is mainly dataset construction, not methodological.

5. Writing tends to overstate contributions (“first real-world benchmark”) and lacks comparison baselines with prior real-scene datasets

**Questions:**

1. How does MathReal differ concretely from MathScape dataset (the one cited above)?
2. Were all images user-captured (not crawled or post-processed)? What is the geographic or device diversity?

---

### Official Review · Reviewer_EQ1P · 2025-10-31

**Soundness:** 2
**Presentation:** 1
**Contribution:** 2
**Rating:** 2
**Confidence:** 3

**Summary:**

This paper introduces MATHREAL, a new benchmark dataset designed to evaluate the mathematical reasoning capabilities of Multimodal Large Language Models (MLLMs) in real-world scenarios. The core contribution is a meticulously curated dataset of 2,000 K-12 math problems, where each problem is an image captured by a handheld mobile device, reflecting authentic usage. The authors introduce a detailed taxonomy of real-world visual challenges, classifying them into 14 subcategories under image quality degradation, perspective variation, and content interference. Through extensive experiments on 40 MLLMs, the paper demonstrates that even state-of-the-art models' performance degrades significantly on these "in-the-wild" images compared to clean benchmarks, with the best model achieving only 53.9% accuracy. The work provides a thorough analysis of model performance, error patterns, and the gap between visual perception and reasoning, highlighting critical areas for future improvement.

**Strengths:**

*   **Originality and Significance:** The paper's primary strength lies in its novel contribution of a "real-world" benchmark. While numerous multimodal math benchmarks exist (e.g., MathVista, MathVerse), they predominantly use clean, synthetic, or post-processed images.
*   **Quality:** The authors employ a rigorous multi-stage process involving: (1) automated and multi-model (GPT-4o, Doubao, Qwen) filtering to ensure data relevance; (2) a three-stage, fully manual annotation process on a dedicated platform; and (3) double-annotation with a third expert for arbitration. This meticulous approach ensures high data quality and reliability. Furthermore, the development of a 14-subtype taxonomy for real-world challenges is a valuable contribution in itself, providing a structured framework for analyzing model failure modes.

**Weaknesses:**

*   **Dataset Language and Generalizability:** A significant limitation, which is only mentioned deep in the appendix (Section C.2, line 913), is that all questions are in Chinese. This should be stated clearly in the abstract and introduction. While a high-quality Chinese benchmark is valuable, this linguistic constraint limits the dataset's immediate utility as a general, global benchmark for evaluating MLLMs, many of which have an English-centric pre-training corpus. The performance of these models may be confounded by language-specific OCR challenges in addition to the visual distortions.
*   **Dataset Scale:** While the quality is high, a dataset of 2,000 instances is relatively small for a benchmark, especially when subdivided across five knowledge categories, three question types, three difficulty levels, and 14 real-world challenge types. The statistical power for analyzing the impact of less frequent challenge subcategories might be limited. This is a common trade-off for high-quality manual annotation, but it's a weakness that should be acknowledged, perhaps with a roadmap for future expansion.
*   **Depth of "Real vs. Clean" Analysis:** The finding that some models (Doubao-1.5-thinking-vision-pro, Gemini-2.5-flash-thinking) perform *better* on real images than clean ones is fascinating and counter-intuitive. The paper briefly speculates this is due to training on authentic data. This point is a major insight and deserves a much deeper analysis rather than a brief mention. What specific "real-world cues" (e.g., paper texture, subtle shadows) might these models be exploiting? Are there architectural commonalities between these models? This could be a small study in itself and would significantly strengthen the paper.
* Some citation formats are not correct.

**Questions:**

1.  **Language and Cross-Lingual Evaluation:** Given that the dataset is in Chinese, could the authors comment on how this might specifically impact the performance of models like GPT-4o or Claude, which are predominantly trained on English data? Do you believe the performance drop is due to a combination of visual noise *and* non-native language OCR, or primarily the visual challenges? It would be crucial to mention this limitation in the main paper. Are there any plans for a multilingual version of MATHREAL?
2.  **Data Distribution Across Challenge Subtypes:** With 2,000 samples distributed across 14 fine-grained subcategories of real-world challenges, what is the sample count for each subtype? Is there sufficient data to draw statistically robust conclusions about the impact of each individual challenge (e.g., "glare" or "reverse-side content") on model performance?
3.  **Probing the "Real > Clean" Phenomenon:** The result where some models perform better on real images is one of the most interesting findings. Could you elaborate on your hypothesis that this is due to training on authentic mobile-captured data? Is it possible to perform a more targeted analysis, for instance, by correlating performance with specific real-world artifacts (e.g., lighting, paper texture) that are absent in the clean versions?
4.  **Disentangling Reasoning from Figure Description:** The text-only LLM baselines were evaluated on `QG + DG` (ground-truth question + ground-truth description). This provides the model with a near-perfect understanding of the visual elements. What is the performance when using only `QG`? This ablation would help quantify the pure reasoning ability of LLMs on these problems when the visual context is entirely absent, providing a clearer baseline for the "reasoning" part of the task.

---

### Official Review · Reviewer_rST1 · 2025-11-01

**Soundness:** 3
**Presentation:** 2
**Contribution:** 3
**Rating:** 6
**Confidence:** 3

**Summary:**

This paper introduces a dataset of 2000 images of math questions paired with textual answers. The images are K-12 math students’ photos of math problems, taken with handheld mobile devices, reflecting how real students may capture physical-world (e.g. textbook or paper) math content. The authors benchmark multilmodal large language models’ (MLLMs’) math question answering abilities on these images.

Not particularly a strength nor a weakness, but this paper follows the general structure of a dataset/benchmark paper: it introduces a benchmark, evaluates models, and then analyzes models’ errors. I do think this paper would be a valueable contribution to the current ecosystem of open benchmarks.

**Strengths:**

It is nice that a portion of this dataset includes pairs of realistic images and cleaned up images, to help researchers better pinpoint weaknesses of MLLMs in a somewhat controlled manner. I also liked the inclusion of metadata around fine-grained subtypes of image noise (e.g. blur, rotation), and annotator-written descriptions of math problem figures (so that the content of this benchmark can also be evaluated in an image-less, text-only manner). The authors also provide really extensive benchmarking results.

**Weaknesses:**

I would have liked to see more details around the provenance of this dataset. I know that exact provenance may be difficult to give due to anonymity contraints, but a few details are missing. For instance, where are these students from (country, region, how many schools)? What languages are present in the data; Figure 1 suggests math images contain Chinese but QA is in English, while Figure 3 says the data is bilingual, but doesn’t specify language? (And if the data is bilingual, does a crosslingual setting affect models vs a monolingual one?)

There are a lot of evaluation results and some of its organization isn’t intuitive. For example, the authors introduce “six experimental settings” at some point but the results aren’t really organized around six experiments. For instance, the results in section 3.3 is organized around a series of hypotheses around what factors may affect models’ performance, and table 2 seems to be organized based on model type: LLMs, open MLLMs, and closed MLLMs. Some more parallel structure between sections could help the reader.

Though extensive benchmarking is good for providing a snapshot of the current model landscape, too much space dedicated to such benchmarking can also shorten the longevity of a paper’s conclusions. I understand that this paper’s main longevity factor is that it presents a challenging benchmark that can be reused for new models. Still, the authors could consider revising Section 3.3 to better forefront model-agnostic takeaways around model performance. This could just be a matter of revisiting sentence order in some paragraphs, or summarizing models’ trends and behaviors with a visual rather than a list of specific models’ results.

**Questions:**

It seems like you had access to data for more subject areas than just math. Why did you decide to only focus on math for this dataset/benchmark contribution? I know data annotation is very expensive but I wonder if it would be good in the future to consider creating a dataset that is similar in size but more diverse in topical area to gain better coverage of different possible model abilities.

How did you get permission to release this data, and from whom did you get permission? Were there any potential concerns around PII (personally identifiable information) in images?

A minor nitpick is that this paper uses the phrasing “real world” a lot, but this term is a bit vague because nearly all data exists in the real world (what does it mean for a dataset to not be real??). Maybe you mean “naturalistic” or “realistic”?

Potentially interesting for the authors is DrawEduMath by Baral et al. 2025 and MathCog by Jin et al. 2025, which are both datasets of images obtained from real K-12 math students.

---

### Note · Authors · 2026-01-06

I have read and agree with the venue's withdrawal policy on behalf of myself and my co-authors.